# Development and Application of Intelligent Assessment System for Metacognition in Learning Mathematics among Junior High School Students

Guangming Wang [ID], Yueyuan Kang *, Zicong Jiao, Xia Chen, Yiming Zhen *[ID], Dongli Zhang and Mingyu Su

Faculty of Education, Tianjin Normal University, Tianjin 300387, China; bd690310@163.com (G.W.); jiaozicong2020@163.com (Z.J.); bxchenxia@163.com (X.C.); zdl921126@163.com (D.Z.); sumy1996@foxmail.com (M.S.)
* Correspondence: kyymail@163.com (Y.K.); zym961224@163.com (Y.Z.)

**Abstract:** Metacognition is one of the key factors that determine students' mathematics learning and affects students' sustainable development. Metacognition assessment has attracted more and more attention from researchers, but how to effectively assess and improve students' metacognition is still unknown. Based on the theoretical basis and practical verification, a mathematics metacognitive intelligence assessment and strategy implementation system for middle school students was developed from both qualitative and quantitative perspectives. This system features the mix of an assessment structural model, assessment scales, a set of norms, improvement strategies and the intelligent assessment and strategy implementation program, which can intelligently output students' mathematical metacognition level and propose targeted improvement strategies. Through the application of the system to 2100 students in Tianjin, China, the results show that the subjects have advantages in mathematical metacognitive knowledge and mathematical metacognitive management. The mathematical metacognitive experience needs to be improved. After intervening with the subjects, according to the improvement strategy provided by the system, it was found that their mathematical metacognition was improved, indicating that the system has a good effect.

**Keywords:** mathematical metacognition; intelligent assessment and policy system; improvement strategies; high school students; norm; mathematics achievement; interview

## 1. Introduction

PISA results show that Chinese students excel in mathematics, but Chinese students study the longest among all participating countries [1]. To a certain extent, this reflects the heavy learning burden and low learning efficiency of Chinese primary and secondary school students.

Metacognition is the knowledge or cognitive activity that reflects or regulates any aspect of cognitive activity and is the core of cognitive activity [2]. Mathematical metacognition is the cognition of one's own mathematical cognition, and its essence is self-awareness, self-monitoring and self-regulation of mathematical cognitive activities, which is an essential factor affecting mathematics learning [3]. Research on the measurement and intervention of mathematical metacognition can help awaken students' metacognitive awareness, improve mathematics learning efficiency, reduce learning burden, thereby enabling students to learn how to learn, and promoting the sustainability of students' learning and development [4–6].

With the continuous development of research on mathematical metacognition, a series of influential mathematical metacognition assessment tools have emerged [7–10]. However, most of these assessment tools can only be used through paper-and-pencil tests or online questionnaires. Researchers often spend a long time collecting data, analyzing data and writing diagnostic reports. This work not only requires a lot of manpower

and material resources but also requires the researcher to have the ability to analyze the data. Additionally, applied research using relevant assessment tools has also emerged [11]. However, most of it only proposes general improvement strategies based on the evaluation results. There are few personalized improvement strategies for students with different metacognitive levels, and there are few intelligent assessment systems.

In the 21st century, the assessment methods for mathematics education are constantly evolving. Marcel and colleagues [12] pointed out that intelligent diagnostic tools are more accurate and efficient than traditional assessment tools for the assessment of mathematical metacognition. Therefore, it is particularly necessary to develop an intelligent assessment system for metacognition in learning mathematics to implement and apply it in junior high schools, and then, to test the effectiveness of the application of the system. In this study, we aim to develop an "intelligent assessment system and strategy implementation for mathematical metacognition for junior high school students" to intelligently analyze the students' questionnaire data. Firstly, it can output thousands of personalized mathematical metacognitive diagnosis reports in a short period of time, which will greatly improve the efficiency of metacognitive assessment and ensure timely and objective feedback on the assessment results, making the assessment process more efficient. Secondly, based on the test results of each student, it can propose a personalized mathematics metacognition improvement strategy, making the mathematics metacognition intervention work more accurately and effectively.

Next, on the basis of the literature review and theoretical framework, the research methods and results will be introduced in detail, and the conclusion based on the findings will also be presented.

## 2. Literature Review

### 2.1. Metacognition

The concept "metacognition" has been introduced in the classic papers of Flavell [2,13,14]. Flavell [13] defined metacognition as "one's knowledge concerning one's own cognitive processes and products or anything related to them" and the active monitoring and consequent regulation and orchestration of these processes (p. 232). According to Flavell, "jotting down key points in a textbook, recording learning while listening to a lecture, preparing a list for supermarket shopping, reading the instructions of an article and feeling puzzled, and repeating a passage in a book that is more difficult to understand are all examples of what is involved in metacognition". He argues that metacognition has two main components. One is metacognitive knowledge, and the other is metacognitive monitoring and self-regulation. Metacognitive knowledge is the individual's knowledge about the influence of cognitive subjects, cognitive tasks, goals, activities, experiences and other factors on cognitive activity. Metacognitive experience is the cognitive or affective experience that accompanies and is subordinate to intellectual activity.

Brown [15,16] distinguished metacognition into two types: knowledge about one's own cognitive activities and regulation of one's own cognitive activities. Knowledge about cognition refers to an individual's knowledge about his or her cognitive resources and his or her compatibility with the cognitive situation. Cognitive regulation refers to the regulation mechanisms used by individuals in the process of problem solving, which includes a range of regulation skills, such as planning, monitoring and evaluation. Dong [6] classifies metacognition into metacognitive knowledge, metacognitive experience and metacognitive monitoring, with an additional component of "metacognitive experience" compared to Flavell and Brown's two categories.

Children begin to develop metacognition in preschool or early school years and reach maturity at about 12 years of age [17,18], metacognition playing an essential role for teenagers in making plans and self-regulation [19]. Lingel and colleagues defined metacognition as the knowledge about cognitive tasks and about cognitive strategies for completing such tasks, including executive skills related to monitoring and self-regulation of one's own cognitive activities [8]. Self-regulation included planning, directing and

evaluating one's behavior. Shilo and Kramarski [9] referred to metacognition as one's self-knowledge of the cognitive processes, which is essential for understanding the task and coming up with solution strategies, and as second-order cognition and thoughts about thoughts and knowledge about knowledge or reflections about actions in line with Flavell [2] and Zohar [20]. The *Encyclopedia of Mathematics Education Second ed*, published in 2020, includes the term metacognition, which is defined "as any knowledge or cognitive activity that takes as its object, or monitors, or regulates any aspect of cognitive activity; that is, knowledge about, and thinking about, one's own thinking" [3].

Based on the above results, most researchers have studied metacognition and its component structure at both qualitative and quantitative levels, and although their views differ slightly, they have basically reached the following consensus: metacognition is people's knowledge of cognitive activities, which is essentially their self-awareness and self-regulation of cognitive activities, and its structural components include metacognitive knowledge, metacognitive experience, metacognitive regulation.

### 2.2. Relationships between Metacognition and Mathematics Performance

It is widely recognized in mathematics education that mathematical metacognition plays an important role in the cognitive process of mathematics, mainly in mathematical problem solving, mathematical thinking and mathematical academic performance [5,6,17]. Several studies have shown that mathematical metacognition has a profound impact on academic performance in mathematics, and there is a positive correlation (r = 0.28) of metacognition with academic performance [6]. The higher the students' mathematics metacognitive level and metacognitive skills, the better their academic performance [21,22].

In line with Verschafel et al. [11], metacognition plays a significant role in mathematics performance [23]. Ohtani and Hisasaka [6] believe that the relationship between metacognition and mathematics is adjusted by the selection of the measurement tool. Online tools in particular have a strong correlation with academic performance, although the correlation between off-line methods and academic performance was weak. Gascoine et al. [21] in their systematic review showed that 61% of studies on children aged 4–16 years used self-report measures. Schneider and Artelt [22] noted that all learners do benefit from metacognitive teaching procedures. De Boer et al. [4] also confirmed the long-term positive effects of metacognitive strategy instruction on students' academic performance, which is consistent with the findings of Dennis and colleagues [24]. Through their study, Desoete et al. [5] found that metacognitive skills in Belgian children were associated with mathematical accuracy throughout elementary school. Metacognitive knowledge was an important predictor of mathematical performance for secondary school students [25]. Baten [26] in a subsequent study confirmed this relationship. In addition, Hacker et al. [27] highlighted an improvement in the accuracy of fraction calculations and of the confidence in those calculations through a metacognitive intervention, within which children become self-regulated learners.

To conclude, metacognition positively correlates with mathematics performance. Therefore, change in mathematics performance is a good indicator to characterize the effect of metacognition improvement.

### 2.3. Assessment of Mathematical Metacognition

A large number of different measures have been used to assess mathematics metacognition. Lingel et al. [8] and Zhao et al. [10] used off-line measures to assess declarative metacognitive knowledge and self-regulatory activities. Online measures were used by Ader [7], Shilo and Kramarski [9] to tap procedural metacognitive knowledge during mathematics performance. Veenman and van Cleef [12] compared online and off-line measures, pointing out the highly convergent validity of the online measures. Veenman and colleagues added that online instruments (especially think-aloud protocols) are to be favored over off-line instruments for the assessment of metacognitive mathematics skills, according to their data [12]. Lingel and colleagues [8] even assumed that different tools

might measure different constructs. It is concluded that the above mathematical metacognitive assessment tools are mainly questionnaire scales, but few artificial intelligence tools are used to implement the assessment. The questionnaire method can be efficient and fast in obtaining research results, but it is a static assessment method, which has disadvantages, such as single content and poor flexibility. Therefore, the questionnaire method often needs to be supplemented together with interviews to dynamically and comprehensively measure changes in individual students' metacognitive levels. This work requires considerable human, material and time resources to complete. Secondly, there is still no "gold standard" for the assessment of metacognition, making study outcomes difficult to compare. In addition, each of these instruments has its own characteristics and cannot be directly used to assess the mathematical metacognition level of Chinese secondary school students without revision.

In the mathematics metacognitive assessment in the Chinese context, Cui et al. [28] developed the Questionnaire on the Metacognitive Levels of Middle School Students in Mathematics. The questionnaire fits the metacognitive characteristics of Chinese middle school students and is a credible and valid instrument for measuring the metacognitive level of Chinese middle school students in mathematics. Based on the developed mathematics metacognitive level questionnaires applicable to different school levels, Wang et al. [29] established the Tianjin metacognitive level normative model of mathematics for middle school students and developed detailed level rating criteria and instructions for using the questionnaire tool, providing a yardstick for assessing the mathematics metacognition of middle school students. Therefore, based on the same research background and research theme, this study will use their mathematics metacognitive questionnaire and norm for further research.

Based on the above studies, there are currently some widely used metacognitive scales. However, there is no corresponding applied research, and there is a lack of proven effective strategies for enhancing metacognition. Therefore, it is urgently required to explore the combination of questionnaires, normative models and artificial intelligence to develop an intelligent assessment system for metacognition in learning mathematics among junior high school students and to innovate the tools for assessment, diagnosis and intelligent administration.

### 2.4. Research Questions

Based on the above review, although there have been many studies on metacognitive concepts, classification types, structural features and assessment, the research on the metacognitive assessment of junior high school students in mathematics is not systematic enough. There is still a lack of representative regional normative models, a lack of research on strategies to improve metacognition, and intelligent assessment improvement systems are rarely seen.

Therefore, the aim of this study is to develop a metacognitive intelligence assessment system for junior high school students in mathematics to implement and apply it in Tianjin, China, and then, to test the effectiveness of the application of the system. The research questions identified for this study include two main areas:

- How to construct a metacognitive intelligence assessment system for junior secondary school students in mathematics?
- How to apply the system in Hexi District, Tianjin City, China, and test the effectiveness of its application?

### 3. Method

#### 3.1. Design and Participants

3.1.1. Designing the Research of the Intelligent Assessment and Strategy Implementation System for Students' Metacognition in Junior High School Mathematics

Based on the existing assessment scales and norms for metacognition among junior high school students in mathematics, this paper formulated a set of strategies aiming

at improving junior high school students' metacognition in learning mathematics. This further led to the development of an intelligent metacognition assessment and strategy implementation program that incorporates all the aforementioned assessment scales, norms and strategies.

Relevant metacognitive improvement strategies are indeed the solution to the crucial issue, which is the first stated research problem. Having analyzed the relevant literature and studied the theoretical foundations, a research case [29] addressing the metacognitive characteristics of high-efficiency mathematics learning among middle school students was used as a reference for the subsequent expert consultation and the formulation of metacognitive improvement strategies for mathematics. The specific research process, as illustrated in Figure 1, is laid out in the following order:

(1) A strategic framework was preliminarily constructed according to the existing metacognition-related research and theoretical foundations;

(2) Greater details were added into the strategies via case-by-case analysis, interviews, etc., after taking into account the aforementioned research on metacognitive characteristics of highly productive middle school students;

(3) The improvement strategies were revised and perfected via expert consultation;

(4) The experts were re-consulted to confirm the strategies;

(5) These expert-approved strategies were embedded into the intelligent strategy implementation module;

(6) The metacognitive improvement strategies were implemented; and lastly,

(7) The strategies' effectiveness was tested.

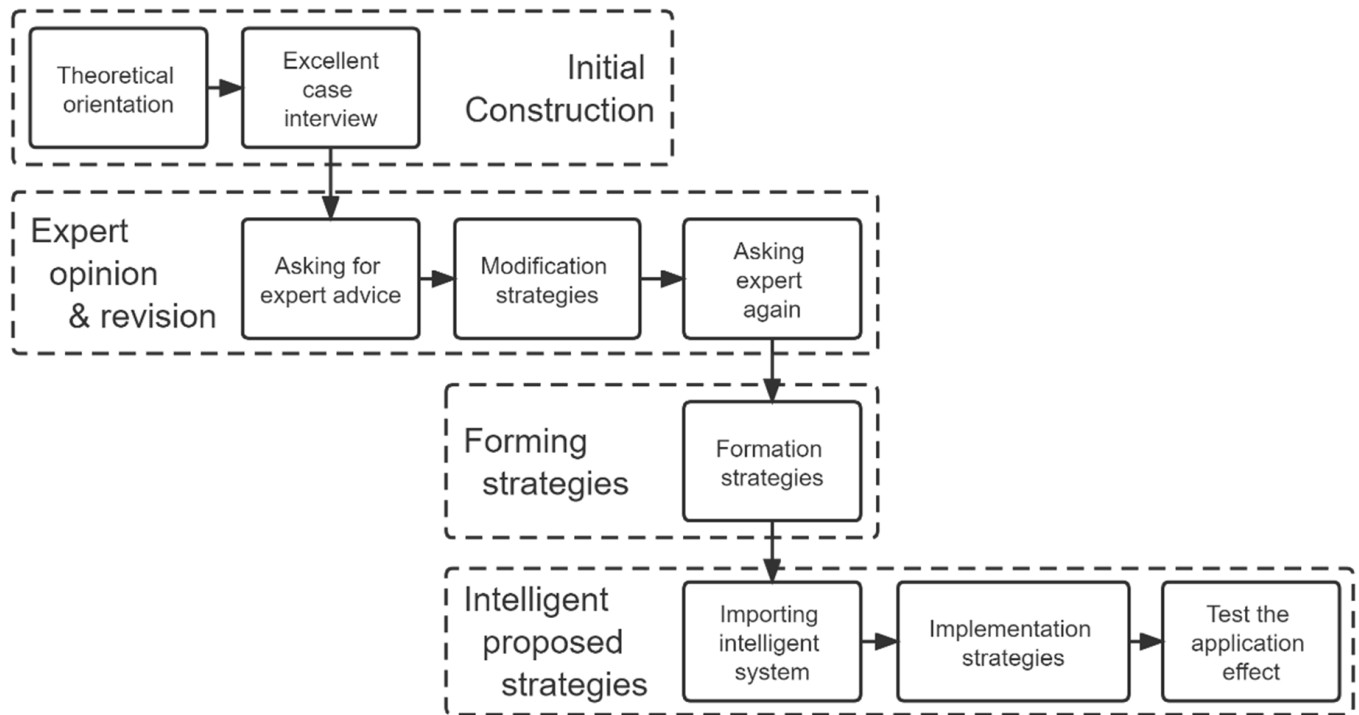

**Figure 1.** Train of thought used in formulating the metacognitive improvement strategies for junior high school mathematics students.

The entirety of this process—from the initial construction of the assessment indicator system to its perfection, and eventually, testing—has resulted in scientific and logically sound "metacognitive improvement strategies for junior high school mathematics", as shown in Figure 1.

3.1.2. Designing the Application Effectiveness Testing and Research of the Intelligent Assessment and Strategy Implementation System

The format adopted for mid-semester and end-of-semester exams organized in all administrative districts across the city of Tianjin, China, is a city-wide unified exam content outline. In order to ensure the consistency of the exam papers received by the students for the academic performance assessment in this research, the scope of the application effectiveness testing for the proposed intelligent assessment module was narrowed down to only the district of Hexi in Tianjin.

Moreover, Tianjin is known to boast considerably high standards in education and economy. With the promulgation of policies advocating educational equity and admission based on geographical proximity, school and student quality differences throughout the city are relatively low, which calls for the adoption of the random sampling method. As a matter of fact, not only have the grade-8 students (second out of the three-year junior high school) mastered a good portion of junior high mathematics, but they also have a relative low pressure in terms of being promoted to the next grade. Putting these advantages into consideration, this study included a total of 2409 grade-8 students from 10 schools in the Hexi District into the questionnaire survey, which features combined online and off-line intelligent testing. In total, 2100 valid questionnaires (1071 boys, 1023 girls, 6 missing data) were returned, showing an 87.17% recovery rate.

Case-by-case intervention was employed according to the respective outcomes of the intelligent assessment and strategy implementation, targeting the general and individual performance. To begin with, 568 tested students (281 boys and 287 girls) from 21 classes across 3 schools were randomly selected for the overall intervention, wherein their metacognitive deficiencies in mathematics were addressed using the improvement strategies suggested by the intelligent assessment and strategy implementation system. The metacognitive deficiencies were indicated by the mathematics test scores, duration needed to study mathematics and pressure felt in learning mathematics; the test papers were graded by the respective schools, while the latter two were evaluated using the corresponding generalized levels in the school of concern as references. As the research progressed, there were six students who obtained sub-par scores in the recent few mathematics tests, took longer than expected studying mathematics, felt pressurized studying mathematics, as well as gained relatively weak understanding of metacognition, metacognitive experience and monitoring of metacognitive performance. With the help of their respective mathematics teachers, three out of these six were given intensive individual interventions using the improvement strategies recommended by the proposed system, and the remaining three were used as controls.

To assess the application effectiveness of the intelligent assessment system, a combination of qualitative and quantitative methods was adopted, focusing specifically on the overall quantitative improvements shown by the big data, as well as on the case-by-case qualitative improvements. On the one hand, to test the overall intervention outcome, this paper first collected the pre- and post-intervention scores for the end-of-semester unified city-wide mathematics examination and evaluated the improvement level based on these score differences. The relevant mathematics teachers were then interviewed at the end of the intervention, with the view of observing the intervention outcomes from an outsider's point of view.

On the other hand, to test the outcome of the individual interventions, both the quantitative and qualitative improvements contributed by the proposed system were also taken into account. Specifically, the differences in metacognitive performance scores before and after the individual interventions were used in conjunction with the students' behavioral manifestations, descriptions of learning productivity, evaluations by self and others, as well as various other qualitative observations.

*3.2. Instruments*

3.2.1. Questionnaire on Metacognition in Learning Mathematics

The Questionnaire of Junior High School Students' Mathematics Metacognition Level developed by Cui et al. [28] was adopted herein to test the students' metacognition in learning mathematics. This questionnaire focused on Chinese middle school students and was developed based on a nationwide test in China, which is consistent with our research background. In addition, some of the questions in their questionnaire involve mathematical knowledge, which can be used to evaluate students' mathematical metacognition in a relatively comprehensive way, which is subject specific and better applicable to this study. The questionnaire contains 36 questions, including 31 questions addressing mathematics metacognition and 5 lie detection questions. Moreover, out of the 36, 11 touch upon mathematical metacognitive knowledge, 7 upon the mathematical metacognitive experience and 13 upon the mathematical metacognitive monitoring (see Appendix A for details).

The overall Cronbach's alpha of the questionnaire is 0.944, while the 0.080 in RMSEA and the above 0.95 NFI, RFI, IFI, TLI and CFI give a good representation of the questionnaire's structural validity and high structural stability. Furthermore, S-CVI/UA (the Scale-level Content Validity Index Universal Agreement, S-CVI/UA = A/m, where A is the number of indices with an expert score of 3 or 4 according to the 4-point grading scale; m is the total number of indices; when S-CVI/UA $\geq$ 0.8, the content validity of the index is high.) and I-CVI (the mean value of Item-level Content Validity Index, I-CVI = B/n, where B is the number of experts giving the score(s) of 3 or 4 according to the 4-point grading scale; n is the total number of experts; when I-CVI $\geq$ 0.9, the content validity of the index is high.) are 0.81 and 0.95, respectively, indicating the questionnaire's above-standard content validity.

3.2.2. Test of Academic Performance in Mathematics

To assess the effectiveness of the improvement strategy, students' scores from two end-of-term mathematics exams were collected, including the second semester of grade 7 and the first semester of grade 8. On the one hand, the quality of the test papers was guaranteed because the test papers were formulated by the Tianjin Education Bureau with the authority (city-wide uniform papers). On the other hand, the test questions can reflect the students' learning of mathematics knowledge because the content of the test paper was the mathematics knowledge that the students of this grade are required to master according to the Chinese Middle School Mathematics Curriculum Standards. Additionally, the authoritative opinions of Mr. Wang Lidong on the quality of exam questions used in this academic performance test were consulted. Mr. Wang, who is working for the China Basic Education Quality Monitoring Collaborative Innovation Center, also set the questions for China's basic education quality monitoring examination for mathematics-related disciplines, as well as international examinations, such as PISA. Mr. Wang gave a relatively high evaluation of the mathematics examination questions used for this test, which suggests the high quality of the examination papers.

3.2.3. Intelligent Assessment System

The intelligent metacognition assessment model for junior high school mathematics students is primarily composed of assessment models (e.g., structural model and assessment indicator system), the assessment scale, regional norms, targeted improvement strategies for students at different levels and the intelligent assessment program, which presents all the above via an "intelligentized" user interface. The specific R&D process is as follows.

**4. Results**

*4.1. Development of Intelligent Assessment System*

The intelligent assessment system for mathematical metacognition of junior high school students mainly included assessment models (structural model, assessment index

system), assessment scale, regional norms, targeted improvement strategies for students at different levels and intelligent assessment software that intelligently presented the above module contents (see Figure 2).

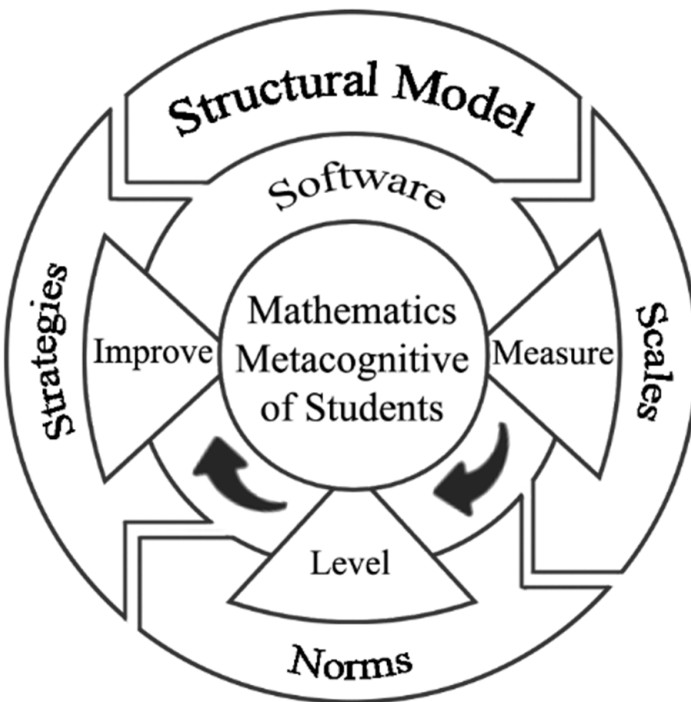

**Figure 2.** Intelligent assessment system and strategy implementation for mathematical metacognition of junior high school students.

Since 2012, our research team has started to develop a series of mathematics learning quality measurement tools for metacognition, non-intelligence factors and learning strategy of primary, junior and senior high school students. Additionally, in practice, regional norms of mathematics learning quality, such as metacognition of students in multiple districts in Tianjin and different studying periods, were developed using the measurement tools. Thirdly, based on the assessment and diagnosis results, personalized improvement strategies were developed and provided for students at different levels of metacognitive awareness. Finally, the intelligent assessment software was developed to compare assessment, diagnosis and regional norms, as well as provide improvement strategies, which were realized intelligently and efficiently using computer software. The above four contents together constituted the intelligent assessment system for mathematical metacognition of junior high school students. The specific research and development process and main content of each module in the system are as follows.

4.1.1. Assessment Theoretical Framework and Assessment Scale

This study was carried out using the Questionnaire of Junior High School Students' Mathematics Metacognition Level developed by Cui et al. [28] (see Appendix A). Firstly, based on the existing metacognition scales [2,30,31], this questionnaire set up questions of junior high school mathematical knowledge, which had certain pertinence and could investigate the mathematical metacognitive level of junior high school students. Additionally, the investigation subjects of this questionnaire were Chinese junior high school students, which is consistent with the investigation subjects of this study. The questionnaire included 3 main dimensions and 10 sub-dimensions (see Figure 3): mathematical metacognitive knowledge (knowledge about individuals, knowledge about tasks and knowledge about strategies), mathematical metacognitive experience (cognitive experience and affective experience) and mathematical metacognitive monitoring (planning, regulation, evalua-

tion, inspection and management). Among them, mathematical metacognitive knowledge refers to the individual cognition of mathematical learning activities, processes and results of the self or others, as well as the knowledge related to these elements. Mathematical metacognitive experience is defined as the perceptual experience of mathematical learning objects and the cognitive experience or emotional experience produced during mathematical cognition. Mathematical metacognitive monitoring indicates the active monitoring and regulation of mathematical cognitive activities, so as to achieve the predetermined goal. The questionnaire consisted of 36 questions (including 5 lie detection questions), which were scored using the five-point Likert-type scale, with high reliability and validity.

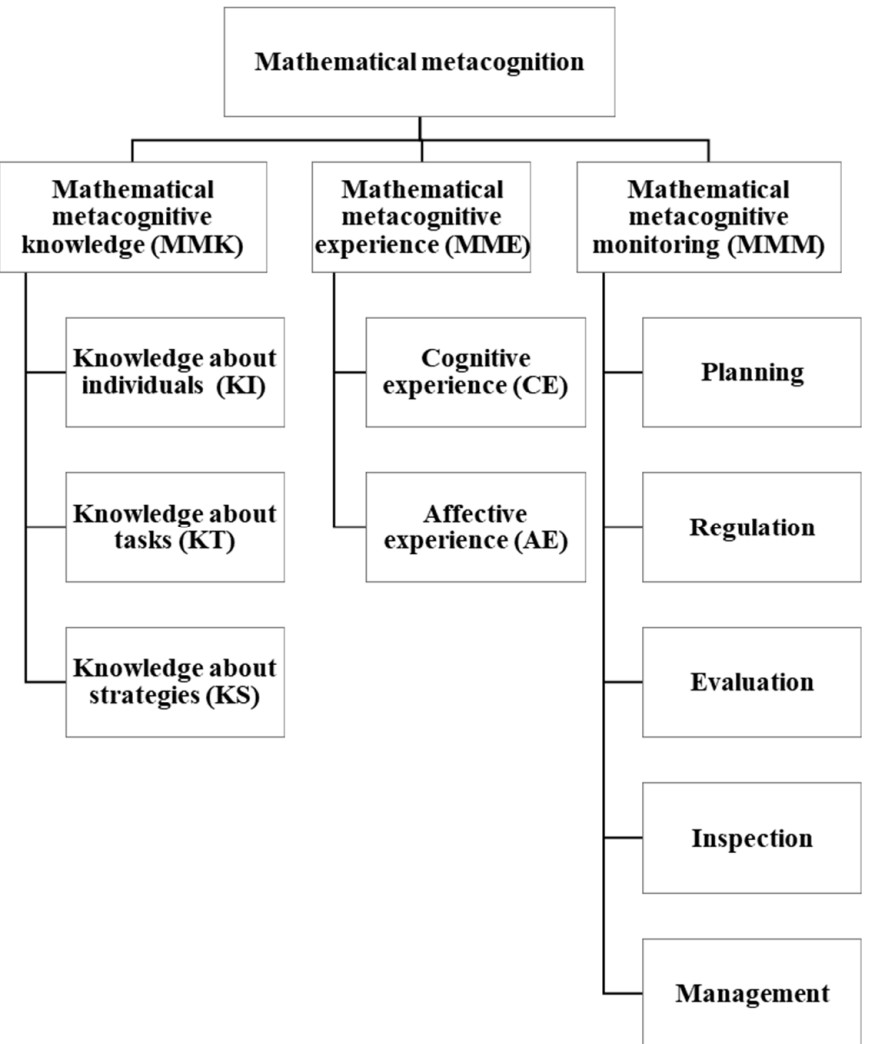

**Figure 3.** Structural model of junior high school mathematical metacognition questionnaire.

4.1.2. Regional Norms

Based on the existing Questionnaire on Mathematical Metacognitive Level of Junior High School Students, the norms of the mathematical metacognitive level of junior high school students in Tianjin were established. Our research team sampled a large number of junior high school students in Tianjin and established regional norms in Tianjin [29]. Two types of norms were selected: percentile rank norms and standard score norms, so that the original scores could be fully explained. The general norms of mathematical metacognition, and the norms of the three main dimensions of metacognitive knowledge, metacognitive experience and metacognitive monitoring and 10 sub-dimensions, were established, and the grade evaluation criteria were also established (see Table 1).

**Table 1.** Grade of mathematical metacognition and its sub-dimensions of junior high school students in Tianjin.

| | Level | T-Score | Raw Score ($X$) | Percentage Rating (PR) |
|---|---|---|---|---|
| Mathematical metacognition | Top | $T \geq 68$ | $X \leq 120$ | $PR \leq 96.58$ |
| | Above Average | $56 \leq T < 68$ | $102 \leq X < 120$ | $73.88 \leq PR < 96.58$ |
| | Average | $44 \leq T < 56$ | $83 \leq X < 102$ | $29.08 \leq PR < 73.88$ |
| | Below Average | $32 \leq T < 44$ | $61 \leq X < 83$ | $3.80 \leq PR < 29.08$ |
| | Low | $T < 32$ | $X < 61$ | $PR < 3.80$ |
| MMK | Top | $T \geq 68$ | $X \geq 45$ | $PR \geq 96.43$ |
| | Above Average | $56 \leq T < 68$ | $38 \leq X < 45$ | $75.25 \leq PR < 96.43$ |
| | Average | $44 \leq T < 56$ | $29 \leq X < 38$ | $31.97 \leq PR < 75.25$ |
| | Below Average | $32 \leq T < 44$ | $19 \leq X < 29$ | $4.02 \leq PR < 31.97$ |
| | Low | $T < 32$ | $X < 19$ | $PR < 4.02$ |
| MME | Top | $T \geq 68$ | $X \geq 31$ | $PR \geq 97.04$ |
| | Above Average | $56 \leq T < 68$ | $24 \leq X < 31$ | $71.83 \leq PR < 97.04$ |
| | Average | $44 \leq T < 56$ | $19 \leq X < 24$ | $32.05 \leq PR < 71.83$ |
| | Below Average | $32 \leq T < 44$ | $14 \leq X < 19$ | $5.85 \leq PR < 32.50$ |
| | Low | $T < 32$ | $X < 14$ | $PR < 5.85$ |
| MMM | Top | $T \geq 68$ | $X \geq 54$ | $PR \geq 96.51$ |
| | Above Average | $56 \leq T < 68$ | $44 \leq X < 54$ | $76.16 \leq PR < 96.51$ |
| | Average | $44 \leq T < 56$ | $34 \leq X < 44$ | $28.63 \leq PR < 76.16$ |
| | Below Average | $32 \leq T < 44$ | $24 \leq X < 34$ | $3.64 \leq PR < 28.63$ |
| | Low | $T < 32$ | $X < 24$ | $PR < 3.64$ |

4.1.3. Development of Improvement Strategy

Initial Construction of Improvement Strategy

(1) Based on theoretical orientation construction of the strategic framework

Research on the cultivation of mathematical metacognitive ability has produced some valuable achievements and conclusions, and its cultivation direction was summarized. Tang et al. pointed out that setting learning objectives and plans, displaying the thinking process using the "think-out-loud method" and consciously reflecting, remedying and summarizing the mathematical learning process can train students' metacognitive ability [32]. Liu believes that the teaching and training strategies of mathematical metacognitive ability mainly include: making students consciously build learning objectives, enhancing students' awareness of others and their own cognitive process, enhancing conscious control of the cognitive process, cultivating and stimulating learning motivation [33]. Stanton et al. [34] have proposed that teachers can cultivate students' metacognitive ability through learning skill guidance, learning process monitoring and group cooperative learning. On the basis of the above research, the improvement strategies were put forward from the 10 sub-dimensions of mathematical metacognition following the training orientation.

(2) Based on excellent case interviews enrichment of specific content of strategy

The assessment tools of this study were used to assess junior high school students in multiple regions, including Beijing and Tianjin in north China, Zhejiang and Shandong in east China, Henan in central China and Qinghai in northwest China. The scope of the assessment was relatively wide, involving regions with different educational levels. According to the assessment and analysis results of each region, 5 students with high mathematical metacognitive levels (30 in total) were selected from each region as interviewees to deeply understand the characteristics of excellent students' mathematical learning habits and methods in the form of e-mail, Wechat and a face-to-face interview. After the interview, the interview contents were summarized, and the interview results were sorted out according to the dimensions as follows:

- Metacognitive knowledge dimension. Excellent students had a strong sense of reflection and would summarize the mastery of mathematical knowledge and the success

or failure of problem-solving results and correctly attribute them. They were good at self-questioning in the problem-solving process and able to extract and transform information in text, graphics and symbols.

- Metacognitive experience dimension. Excellent students had the habit of previewing knowledge in advance and summarizing the knowledge framework independently. They had a more positive emotional experience during mathematical learning and could overcome or transform negative experience.
- Metacognitive monitoring dimension. Excellent students would formulate reasonable learning, problem-solving or examination plans, timely adjust their learning mentality and regularly test and evaluate their mathematical learning results.

Through the above literature analysis and excellent case interviews, combining theoretical and practical foundations, the general content and training direction of mathematical metacognitive training strategies were understood, and specific and operable improvement strategies were preliminarily put forward based on the 10 sub-dimensions of mathematical metacognition (see Appendix B).

Revision of Improvement Strategy

The initially constructed Improvement Strategy for Mathematical Metacognition of Junior High School Students was revised, and the modification opinions were solicited from experts through semi-structured interviews. To ensure expert validity, the background and number of experts were needed to be ensured during expert consultation. In terms of expert background, the experts consulted should have a mathematics background, education background, psychology background and rich education and teaching research experience. Therefore, the improvement strategy was revised based on the opinions of university professors in the field of mathematics education, researchers of the China Basic Education Quality Monitoring Collaborative Innovation Center, mathematics textbook editors and middle school mathematics teachers. As for the number of experts, the expert consultation adopted purposeful sampling, and a total of 10 experts were consulted. Among them, there were four professors from Beijing Normal University, Tianjin Normal University and Qinghai Normal University in the field of mathematics education, one researcher from the China Basic Education Quality Monitoring Collaborative Innovation Center, one middle school mathematics textbook editor from the People's Education Press and four middle school mathematics teachers with theoretical research and practical exploration in metacognition.

After consulting the experts, all experts agreed with the overall design and strategic framework and put forward their opinions on the pertinence and logicality of the strategy. The details are as follows:

- There were repeats in suggestions of each sub-dimension and problems in orientated dimensions. For example, experts pointed out that "both knowledge and regulation dimensions of the strategy paid attention to the application of metacognitive cues, and the strategic orientation dimensions needed to be further considered", "the examination-oriented mentality did not belong to the learning mentality, so whether it should be considered in the dimension of regulation", and "checking writing errors did not belong to the planning dimension".
- The logicality between the recommendations still needed to be strengthened. For instance, the logicality of "first judge whether it conforms to their own situation, and then actively improve their own mathematical learning" should be strengthened. The logical relationship between "paying attention to group honor" and "comparing your past mathematical learning state and achievement with the present" is unclear.
- The language was not refined and accurate enough. Experts suggested that the language should be refined in many parts of the full text. For example, "Facing the praise of teachers and classmates or the progress in achievements" could be changed to "when making progress or being praised". "Pay attention to problem-solving dexterity" is inaccurate and difficult for students to understand and implement. The

overall modification suggestions for the mathematical metacognitive improvement strategy are displayed in Table 2.

**Table 2.** Modification suggestions on improvement strategy for mathematical metacognition of junior high school students.

| Dimension | Sub-Dimension | Modification Suggestion |
|---|---|---|
| MMK | KI | No change is needed |
| | KT | Strengthen logicality and check for typos |
| | KS | Give some examples to help students understand |
| MME | CE | The preview part should focus on the study of algebraic knowledge rather than the solution of algebraic problems |
| | AE | Consider the overall logical structure and adjust the language to highlight the positive function of homework |
| MMM | Planning | Some recommendations do not fall into this category, and refine language |
| | Regulation | When giving advice to students, do not use words that are not easy to understand, such as "dexterity" |
| | Inspection | Deleting part is not a recommendation for test dimension |
| | Evaluation | Do not focus too much on solving problems. Solving problems is not the totality of mathematics |
| | Management | Carefully study the definition of this dimension in scale papers and highlight "management" |

Determination of Improvement Strategy

For the existing problems, the improvement strategy was revised again, and the countermeasures for improving the mathematical metacognition of junior high school students were finally obtained, as seen in Table 3.

**Table 3.** Countermeasures for improving the mathematical metacognition of junior high school students.

| Performance | Improvement Strategy |
|---|---|
| Dimension: Mathematical metacognitive knowledge (55 points) | |
| Sub-dimension: Knowledge about individuals (25 points) | |
| Middle-level students ($13 \leq X < 17$): <br> 1. There is room for improving the understanding of one's own ability and effort level; <br> 2. Perform average in understanding their own knowledge mastery and problem-solving ideas; <br> 3. There is room for improving the understanding of one's own mathematical learning characteristics. <br><br> Low-level students ($X < 13$): <br> 1. The understanding of one's own ability and effort level is deficient; <br> 2. Perform poorly in understanding their own knowledge mastery and problem-solving ideas; <br> 3. There is a poor understanding of one's own mathematical learning characteristics. | 1. Regularly reflect on the situation of mathematical learning and analyze the causes <br><br> Students should regularly reflect on their own grasp of mathematical knowledge and completion of exercises, analyze the reasons for the poor grasp of knowledge or mistakes in exercises, classify the reasons into personal ability, effort level, the difficulty level of task, etc., and treat their own ability and effort level more objectively through reflection. <br><br> 2. Summarize and review the mathematical knowledge frame of each chapter <br><br> After the learning of a chapter is finished, students should sort important theorems and definitions, teachers' supplements and types of error-prone questions into a knowledge frame (dendrogram, list, etc.) and review it in time. During the review, they should recall the knowledge points and the solving ideas of error-prone questions by the knowledge frame, mark the forgotten contents and fully understand their own advantages and disadvantages in knowledge and problem solving. <br><br> 3. Correctly understand the differences in mathematical learning between themselves and their peers <br><br> Communicate with peers to share their own mathematical learning (how to preview, take class, review, etc.), compare the similarities and differences with peers and keep up common and good characteristics; actively improve their own mathematical learning according to a personal situation. |

**Table 3.** *Cont.*

| Performance | Improvement Strategy |
|---|---|
| Sub-dimension: Knowledge about tasks (15 points) | |
| Middle-level students (8 ≤ X < 10):<br><br>1. The understanding of the nature and requirements of mathematical problems remains to be improved;<br>2. There is room for improving the understanding of the types of mathematical tasks and the time needed to complete tasks;<br>3. Perform average in the application of mathematical knowledge.<br><br>Low-level students (X < 8):<br><br>1. Have a poor understanding of the nature and requirements of mathematical problems;<br>2. There are deficiencies in understanding the types of mathematical tasks and the time required to complete tasks;<br>3. Perform poorly in the application of mathematical knowledge. | 1. Remind oneself in the process of solving problems<br><br>Students should remind themselves to consider the error-prone points before solving a problem. For example, facing an equation solving problem, one should consider the denominator is not zero when the fraction value is zero. Additionally, remind oneself to think about the knowledge examined by the problem and the skills used to solve the problem after solving the problem to comprehensively deepen personal understanding of the question.<br><br>2. Scientifically analyze and plan mathematical tasks<br><br>First, for multiple mathematical tasks, students should analyze whether the purpose is to summarize knowledge, consolidate practice methods or improve mathematical ability, determine the task types and complete them one by one from easy to difficult; second, they should estimate the time to complete each task, work out a good plan and strengthen their awareness of efficiency.<br><br>3. Use variants to grasp the key characteristics of mathematical knowledge<br><br>Summarize the application of the same knowledge point in questions about variants, appropriately find some analytical cases to help oneself pay attention to the new characteristics of the same knowledge to promote one's better understanding of knowledge and better transfer and application of knowledge. |
| Sub-dimension: Knowledge about strategies (15 points) | |
| Middle-level students (8 ≤ X < 10):<br><br>1. Perform average in reflecting on the use of mathematical learning strategies;<br>2. There is room for improvement in choosing mathematical ideas and methods;<br>3. Perform average in judging the advantages and disadvantages of problem-solving methods.<br><br>Low-level students (X < 8):<br><br>1. Perform poorly in reflecting on the use of mathematical learning strategies;<br>2. Are weak in choosing mathematical ideas and methods;<br>3. Are weak in judging the advantages and disadvantages of problem-solving methods. | 1. Strengthen the study of problem-solving strategies<br><br>Polya's *How to Solve It* is recommended. According to the four steps set out in Polya's book: figure out the meaning of the question, work out a plan, carry out a plan, review and reflect. Students should reflect on how they read key information, organize the thought and find the relationship between known conditions and unknown variables and what strategies they use in the process.<br><br>2. Summarize mathematical ideas and methods<br><br>In the process of learning concepts and theorems, students should pay attention to the mathematical ideas and methods (conversion, symbolic–graphic combination, structure, etc.) and summarize by themselves every week; in the problem-solving process, they should first identify whether the problem situation is similar, then recall the relevant knowledge points and the mathematical ideas and methods that may be used and remind themselves to use the corresponding strategies.<br><br>3. Employ an explicit thinking process<br><br>Answer questions actively in class and broaden one's methods of solving problems by dictating or writing down the solving process and thinking about others' solving methods. Explain the problem to each other after class and summarize multiple solutions to one problem to improve one's understanding of the advantages and disadvantages of problem-solving methods. |

**Table 3.** *Cont.*

| Performance | Improvement Strategy |
|---|---|
| **Dimension: Mathematical metacognitive experience (35 points)** | |
| **Sub-dimension: Cognitive experience (20 points)** | |
| Middle-level students ($11 \leq X < 14$):<br>1. There is room for improvement in understanding the difficulty levels and familiarity levels of learned mathematical contents;<br>2. Perform average in experiencing and overcoming math learning difficulties.<br>Low-level students ($X < 11$):<br>1. Are weak in understanding the difficulty levels and familiarity levels of learned mathematical contents;<br>2. There are deficiencies in experiencing and overcoming math learning difficulties. | 1. Preview before class<br>Preview before class, highlight concepts, try to solve examples, find and mark difficult and unfamiliar knowledge and then listen to instruction with questions in class. Through preview, one can feel the connection between old knowledge and new knowledge, review relevant old knowledge and smoothly transit from old knowledge to new knowledge.<br>2. Reflect on the ways of overcoming learning difficulties<br>Recall the difficulties one has encountered in the process of mathematics learning (unable to solve problems; do not understand knowledge points, etc.), reflect on one's ways to overcome difficulties (consult others/carefully analyze and mark what is puzzling, etc.). When encountering similar mathematical difficulties again, calm down to analyze and follow the previous experience of overcoming difficulties to solve problems, improve the ability to solve mathematical difficulties. |
| **Sub-dimension: Affective experience (15 points)** | |
| Middle-level students ($8 \leq X < 10$):<br>1. There is room for improvement in experiencing a sense of achievement in mathematics learning;<br>2. The confidence in math learning remains to be improved.<br>Low-level students ($X < 8$):<br>1. Seldom experience a sense of achievement in mathematics learning;<br>2. Lack confidence in math learning. | 1. Study in pairs to motivate oneself<br>Study in pairs with a classmate with equivalent mathematics level. Set a common goal for each test, compete with each other, change the partner after making progress and set a new goal. Stimulate one's own potential through peer competition, strive for the goal to make progress and experience a pleasant emotion.<br>2. Change one's passive attitude toward homework<br>Treat homework correctly, change the passive attitude of "report on one's mission to the teacher", regard homework as a means to test one's knowledge mastery and application of ideas and methods, ask questions proactively and summarize knowledge points or ideas/methods to experience the joy of gaining knowledge in the process of mastering and applying knowledge.<br>3. Give oneself positive psychological hints<br>When encountering mathematical learning difficulties, one should encourage oneself and tell oneself, "Others can do it. I can, too" or "I can do it" to alleviate the impact of negative emotions on mathematical learning and enhance one's confidence in solving problems. |
| **Dimension: Mathematical metacognitive monitoring (65 points)** | |
| **Sub-dimension: Planning (10 points)** | |

**Table 3.** *Cont.*

| Performance | Improvement Strategy |
|---|---|
| Middle-level students (5 ≤ $X$ < 7):<br>There is room for improvement in working out math learning plans.<br>Low-level students ($X$ < 5):<br>Are weak in working out math learning plans. | 1.    Prepare daily to-do lists and set long-term goals<br><br>Prepare a to-do list every day, including daily tasks and their corresponding time quantum, strictly implement the list, avoid delaying or forgetting any task. Develop long-term mathematical learning goals and decompose them into more operational near-term goals, such as "my weekly test result must exceed that of xxx classmate" to motivate oneself to complete learning tasks.<br><br>2.    Clarify problem-solving plans<br><br>Before answering a question, fully understand the meaning of the question and mark the key information; in the process of solving a problem, find out the implied conditions, formulate a solving plan and reversely infer from the target; after solving a problem, check whether the results are in line with the background of the question and whether there are multiple or missing answers. |
| Sub-dimension: Regulation (20 points) | |
| Middle-level students (11 ≤ $X$ < 14):<br>1.    There is room for improvement in adjusting problem-solving methods;<br>2.    Perform average in adjusting the attitude to mathematics learning;<br>3.    Perform average in adjusting learning steps and plans.<br>Low-level students ($X$ < 11):<br>1.    Perform poorly in adjusting problem-solving methods;<br>2.    Are weak in adjusting the attitude to mathematics learning;<br>3.    Are weak in adjusting learning steps and planning. | 1.    Pay attention to self-questioning to adjust methods<br><br>If there is more than one idea, but one is sure about none, one should encourage oneself to explore each idea a little bit, apply heuristic self-questioning, such as "Are the plan, steps and methods correct? Why?". One can adjust one's thinking process and problem-solving methods through self-awareness.<br><br>2.    Adjust one's mentality in time<br><br>Treat mathematical learning with a usual mind. When one makes progress or is praised, one should be modest and prudent to avoid complacency; when one falls behind or is criticized, one can talk with peers, seek help and relieve one's frustration.<br><br>3.    Adjust mathematics learning steps and plans in time<br><br>Reflect on one's own mathematical learning situation regularly (e.g., problem-solving speed, problem-solving ideas, etc.) to find one's own shortcomings, timely adjust the learning plan to make up for the shortcomings, such as enhancing mastery through practice. |
| Sub-dimension: Evaluation (10 points) | |
| Middle-level students (5 ≤ $X$ < 7):<br>1.    There is room for improvement in evaluating their own mastery of classroom content;<br>2.    Do average in evaluating their own problem-solving thinking.<br>Low-level students ($X$ < 5):<br>1.    Are weak in evaluating their own mastery of classroom content;<br>2.    Are weak in evaluating their own problem-solving thinking. | 1.    Actively use "a question chain" to evaluate<br><br>Actively use "a question chain" to evaluate one's own knowledge mastery., e.g., "What are the knowledge points involved in this class? Which have been mastered? Which have not? Additionally, why? What are the countermeasures?".<br><br>2.    Write reflective weekly reports on mathematics<br><br>Comprehensively reflect on the thinking process of problem solving, conduct multi-angle (problem-solving ideas, problem-solving ways, etc.) observation according to the structural characteristics of problems and write weekly reports on the problem-solving methods acquired through reflection. In the future study, apply the acquired problem-solving methods, reflect on the problems existing in the application, seek ways to overcome, supplement and improve reflective weekly reports to gradually develop the habit of reflection–remediation–reflection. |

**Table 3.** *Cont.*

| Performance | Improvement Strategy |
|---|---|
| Sub-dimension: Inspection (15 points) | |
| | 1.    Mark task lists to test one's own learning results |
| Middle-level students ($8 \leq X < 11$):<br><br>1.    There is room for improvement in testing their own learning results;<br>2.    Do average in judging the rationality of their chosen problem-solving strategy.<br>Low-level students ($X < 8$):<br><br>1.    Perform poorly in testing their own learning results;<br>2.    Are weak in judging the rationality of their chosen problem-solving strategy. | Students mark daily or weekly task lists (divided by star levels), check and evaluate each other with peers and exchange and share experiences; complete after-class exercises in the textbook or related exercises after each class to understand one's own knowledge application situation.<br><br>2.    Self-test to judge the rationality of strategies<br><br>After solving the problems, reflect on the effectiveness of the strategies used and look for similar problems to solve to test the generalizability of the solution strategies. If problems are found in the use of the solution strategy, find the reasons and avoid the unreasonable use of the solution strategy. |
| Sub-dimension: Management (10 points) | |
| | 1.    Take notes and sort out mistakes |
| Middle-level students ($5 \leq X < 7$):<br><br>1.    There is room for improvement in inducing mathematical knowledge;<br>2.    Do average in summarizing mathematical problem-solving methods.<br>Low-level students ($X < 5$):<br><br>1.    Are weak in inducing mathematical knowledge;<br>2.    Are weak in summarizing mathematical problem-solving methods. | Quickly take down the teacher's important blackboard writing and supplementary knowledge in class and complete the notes after class. Sort out one's mistakes according to knowledge points and specify the causes of mistakes, the background knowledge involved and the problem-solving methods used.<br><br>2.    Strengthen the management of problem-solving methods<br><br>When solving mathematical problems, one should use equations, image, forms, etc., to comprehensively analyze the problem information and summarize general methods and skills for common questions., e.g., the solution to set problems about the system of inequalities often involves drawing a number axis. |

Note: Above middle-level are high-level students, and the improvement strategy is "Keep up".

### 4.1.4. Intelligent Assessment Software

The research team independently developed the mathematics metacognitive intelligence assessment and policy software for primary and secondary school students. The software was developed with Microsoft Visual Studio Community 2019 tools and NPOI plugin, combined with SunnyUI for interface beautification. Two intelligent assessment tools were designed and developed for different scenarios, namely, the Individual Student Edition and the Integrated School Edition, which provide reports and recommendations for individual students and schools and districts as a whole. Both software embedded with a complete math metacognition scale and improvement strategies for junior high school students, which can evaluate students professionally and provide professional and directed personalized advice for improvement according to the assessment results. The difference is that the Individual Student Edition software can be self-diagnosed by students, providing diagnosis results and improvement suggestions in real time. Integrated School Edition software takes the school and the region as a whole, and the batch is for the region, school, class as a whole and all the students tested within a short period of time. In turn, the output visual diagnostic results provide suggestions for improvement.

On the one hand, after students complete the test independently using the software, they can view the diagnosis results independently and immediately; at the same time, the system can automatically and immediately feed back improvement measures according to the level of the diagnosis results, which is convenient for students to independently diagnose and view the results anytime, anywhere. Figure 4 is an example of a student's self-assessment of mathematical metacognitive intelligence and immediate feedback on improvement suggestions.

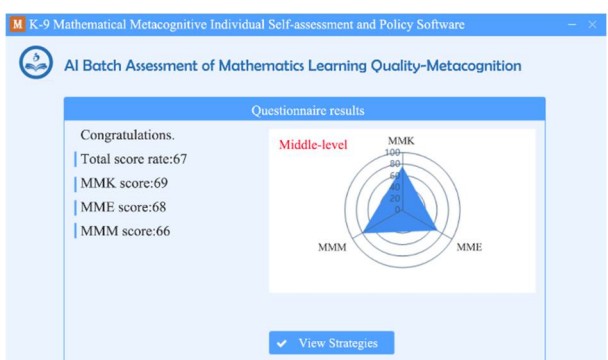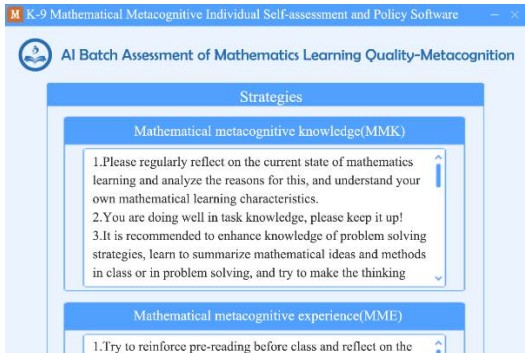

**Figure 4.** Visualized diagnosis results of mathematical metacognitive individual self-assessment and policy software.

On the other hand, the intelligent batch assessment and policy software can automatically calculate the total score of the main dimension and the score of the sub-dimension according to the results filled in by the students. The software was divided into three modules: the first was the basic information collection module; the second was the scale data collection module; and the third was the results and recommendations output module (see Figure 5). Users can view the scores and levels in different interfaces, such as the main dimension, sub-dimension, total score level, sub-dimension scoring rate and status quo performance. The diagnostic results and improvement countermeasure reports of mathematical metacognition can also be automatically exported in batches.

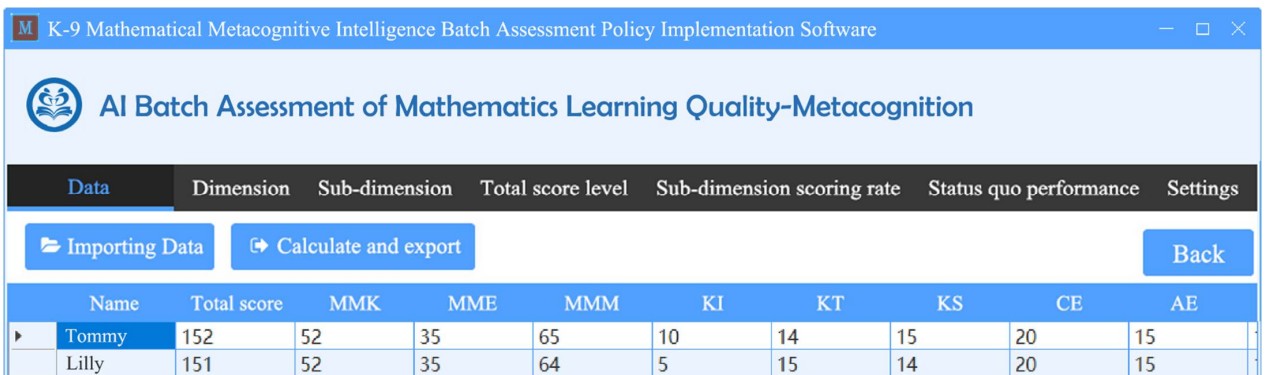

**Figure 5.** The functional interface of the software for mathematics metacognitive intelligence batch assessment and policy implementation.

The software uses technology to accurately measure students' level of mathematical metacognition, enhance students' knowledge of their own level of mathematical metacognition and greatly improve the quality of their mathematical learning.

### 4.2. Intelligence Assessment Diagnosis and Norm-Referenced Analysis

4.2.1. Results of Mathematics Metacognitive Diagnosis of Junior High School Students of Hexi District and Norm-Referenced Analysis

(1) Results of mathematics metacognitive diagnosis of junior high school students of Hexi District

Descriptive statistics of students' mathematics metacognition and the primary dimensions were conducted (see Table 4), yielding an average value of 121.3605 for the questionnaire (full score: 155) and an overall scoring rate (the scoring rate refers to the percentage corresponding to the ratio of the test score divided by the full score—scoring rate = mean/dimension full score. The higher the score rate, the more concentrated it is in the high segment, and the higher the student's mathematical metacognition level)

of 78.30%, indicating generally good mathematics metacognition among the students. Specifically, the highest and lowest scores on the assessment were 155 and 31, respectively. Paired *t*-test comparisons were performed on the three primary dimensions. The results showed that mathematical metacognitive experience was significantly different from the other two primary dimensions ($t_1 = 29.145$, $t_2 = 29.512$, $p = 0.000 < 0.001$), and its effect size fell between medium and large effects ($d_1 = 0.728$, $d_2 = 0.739$), while mathematical metacognitive knowledge had no significant difference from mathematical metacognitive monitoring ($t = 0.563$, $p = 0.574 > 0.05$). Specifically, the scoring rates on mathematical metacognitive knowledge and mathematical metacognitive management were quite close and higher than the overall scoring rate of the questionnaire; comparatively, the scoring rate on mathematical metacognitive experience was relatively low ($S = 71.46\%$)—lower than the overall scoring rate of the questionnaire.

**Table 4.** Results of descriptive statistics of mathematics metacognition and primary dimensions.

| | N | Min | Max | Mean (M) | Standard Deviation | Dimension Full Score | Scoring Rate (S) |
|---|---|---|---|---|---|---|---|
| Mathematical metacognition | 2100 | 31 | 155 | 121.3605 | 18.59534 | 155 | 78.30% |
| MMK | 2100 | 11 | 63 | 44.1105 | 7.56249 | 55 | 80.20% |
| MME | 2100 | 7 | 35 | 25.0095 | 3.49106 | 35 | 71.46% |
| MMM | 2100 | 13 | 75 | 52.2405 | 8.99428 | 65 | 80.37% |

With respect to mathematics metacognitive knowledge, the scoring rates on knowledge about individuals (KI), knowledge about tasks (KT) and knowledge about strategies (KS) on the secondary dimension were 82.14%, 81.99% and 75.19%, respectively (see Table 5), exhibiting a descending pattern. The scoring rates on KI and KT were higher than those of the mathematics metacognitive knowledge on the primary dimension (S = 80.20%), while the scoring rate of KS was lower than that of the metacognitive knowledge.

**Table 5.** Results of descriptive statistics of mathematics metacognition and primary dimensions.

| | N | Min | Max | Mean (M) | Standard Deviation | Dimension Full Score | Scoring Rate (S) |
|---|---|---|---|---|---|---|---|
| KI | 2100 | 5.00 | 25.00 | 20.5338 | 3.53100 | 25 | 82.14% |
| KT | 2100 | 3.00 | 39.00 | 12.2988 | 2.23434 | 15 | 81.99% |
| KS | 2100 | 3.00 | 39.00 | 11.2780 | 2.64080 | 15 | 75.19% |
| CE | 2100 | 4.00 | 20.00 | 14.3033 | 2.28116 | 20 | 71.52% |
| AE | 2100 | 3.00 | 15.00 | 10.7062 | 1.89552 | 15 | 71.37% |
| Planning | 2100 | 2.00 | 36.00 | 7.6064 | 1.92350 | 10 | 76.06% |
| Regulation | 2100 | 4.00 | 20.00 | 16.6011 | 2.80477 | 20 | 83.01% |
| Evaluation | 2100 | 2.00 | 10.00 | 7.8237 | 1.69608 | 10 | 78.24% |
| Inspection | 2100 | 3.00 | 15.00 | 11.6604 | 2.53955 | 15 | 77.74% |
| Management | 2100 | 2.00 | 10.00 | 8.5488 | 1.41231 | 10 | 85.49% |

With respect to mathematics metacognitive experience, the scoring rates of cognitive experience (CE) and affective experience (AE) on the secondary dimension were 71.52% and 71.37%, respectively, which was close to that of the mathematics metacognitive experience on the primary dimension (S = 71.46%).

For mathematics metacognitive monitoring, the scoring rates of planning, regulation, evaluation, inspection and management on the secondary dimension were 76.06%, 83.01%, 78.24%, 77.74% and 85.49%, respectively. Specifically, apart from regulation and management, the other secondary dimensions all had scoring rates lower than that of the mathematics metacognitive monitoring on the primary dimension (S = 80.37%).

Based on the survey of mathematics metacognitive levels of junior high school students at Hexi District, Tianjin, the secondary dimensions are ranked in the following descending order in terms of the scoring rates: management, regulation, KI, KT, evaluation, inspection, KS, CE and AE, indicating that the improvement on the subjects' CE and AE should be focused on.

(2)    Norm-referenced analysis of the levels of students in Hexi District against the norm of Tianjin

To further diagnose the mathematics metacognitive levels of junior high school students in Hexi District, the mathematics metacognitive norm of junior high school students of Tianjin [29] was applied to accurately diagnose the advantages and disadvantages of mathematics metacognition in the former. The calculation shows that the average score of mathematics metacognition in junior high school students in Hexi District was 121.3605, with 97.67 percentile points, indicating they surpass about 97.67% of junior high school students in Tianjin in terms of mathematics metacognitive levels. It can be said that their overall mathematics metacognition is at the top level in the norm of Tianjin.

To accurately diagnose the specific characteristics of the subjects, the norms of the three primary dimensions in Tianjin were used to evaluate subjects' average scores. As can be seen from Figure 6, the subjects' mathematical metacognitive knowledge and mathematical metacognitive monitoring basically reached the top level in the norms, while there is still a considerable gap between their mathematical metacognitive experience and the top level of the norm, indicating the students' mathematical metacognitive experience needs to be improved.

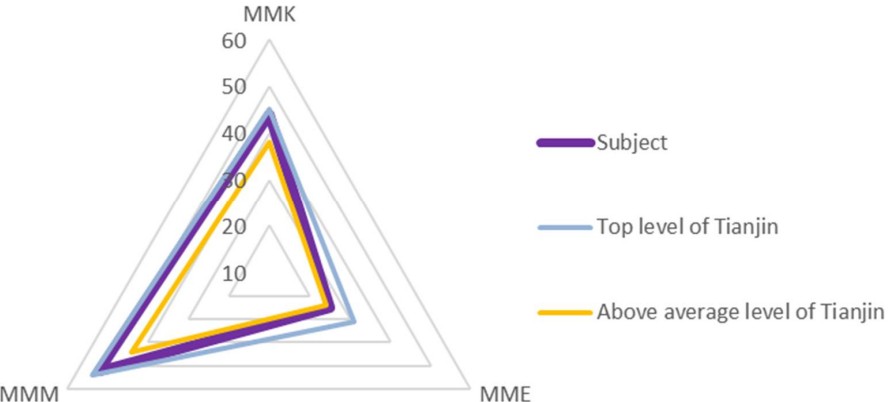

**Figure 6.** Comparison of subjects' primary dimensions of mathematics metacognition with the norms of Tianjin City.

To accurately diagnose the subjects' level of mathematical metacognitive experience, the mathematics metacognitive experience norm (see Table 1) was applied. As can be seen, the average value of subjects' mathematics metacognitive experience is 25.0095, which is at a medium to high level with a percentile point of 78.29, indicating there are about 78.29% of junior high school students in Tianjin with a lower cognitive experience level. The sub-dimensional norm of mathematical metacognitive experience of Tianjin was applied to analyze the limitations of the subjects in detail (see Table 4). The T scores of subjects' average scores on these two dimensions were calculated. It was found that both their cognitive experience and affective experience are somewhere between the medium to high level and top level. Notably, compared with the sub-dimension of cognitive experience, the subjects showed an evident shortfall in affective experience compared with the top level. In general, subjects' cognitive experience and affective experience levels were higher than the overall level of students in Tianjin, indicating that these students are able to obtain a wealth of cognitive and affective experience and have a good mentality and confidence.

As can be seen from the above analysis, junior high students in Hexi District generally have an excellent mathematics metacognitive level and particularly enjoy a significant edge in terms of mathematical metacognitive knowledge and mathematical metacognitive management. Specifically, in the subsequent intervention and improvement actions, attention should be paid to students' mathematical metacognitive experience, and improvements in their cognitive experience and affective experience should be focused on.

### 4.2.2. Results of Mathematics Metacognitive Diagnosis of Students Participating in the Case Study and Norm-Reference Analysis

Based on the diagnostic results of classes in the case study, we consulted H, the head and mathematics teacher for a junior grade-2 class at school *X* and selected six students who had a medium level of mathematics metacognition and suffered long study time and huge study pressure in mathematics. Here, we code them as Student A to Student F to replace their true identity information.

As can be seen in Table 6, the raw mathematics metacognitive scores of these six students were 108, 111, 110, 108, 110 and 108, respectively, with their scoring rates being 69.67%, 71.61%, 70.97%, 69.67%, 70.97% and 69.67%, pointing to a medium level in the norm.

**Table 6.** Results of norm-reference analysis of mathematics metacognition in (6) students participating in the case study.

| Student | Raw Score | T-Score | Norm Level | Scoring Rate (S) |
|---------|-----------|---------|------------|------------------|
| A | 108 | 59.9 | Average | 69.67% |
| B | 111 | 61.3 | Average | 71.61% |
| C | 110 | 60.9 | Average | 70.97% |
| D | 108 | 59.9 | Average | 69.67% |
| E | 110 | 60.9 | Average | 70.97% |
| F | 108 | 59.9 | Average | 69.67% |

In terms of metacognitive knowledge, students A and D are at a medium level, while the remaining students are at a medium to high level. From the perspective of cognitive experience, students C is at a medium to high level, students A, D and F are at a medium level, and students B and E are at a low to medium level. In terms of metacognitive monitoring, students A, B, D and E are at a medium to high level, while students C and F are at a medium level. Therefore, students A and D have relatively weak mathematics metacognitive knowledge; students B and E have relatively weak mathematics metacognitive experience; and students C and F have relatively weak mathematics metacognitive monitoring (see Table 7).

**Table 7.** Results of norm-reference analysis of primary dimensions of mathematics metacognition in (6) students participating in the case study.

| Dimension | Student | Raw Score | T-Score | Norm Level | Scoring Rate (S) |
|-----------|---------|-----------|---------|------------|------------------|
| MMK | A | 34 | 51.9 | Average ($29 \leq X < 38$) | 61.82% |
| | B | 42 | 61.9 | Above average ($38 \leq X < 45$) | 76.36% |
| | C | 43 | 63.5 | Above average ($38 \leq X < 45$) | 78.18% |
| | D | 36 | 54.4 | Average ($29 \leq X < 38$) | 65.45% |
| | E | 40 | 59.5 | Above average ($38 \leq X < 45$) | 72.73% |
| | F | 43 | 62.82 | Above average ($38 \leq X < 45$) | 78.18% |
| MME | A | 21 | 49.2 | Average ($19 \leq X < 24$) | 60.00% |
| | B | 16 | 39.3 | Below average ($14 \leq X < 19$) | 45.71% |
| | C | 24 | 56.1 | Above average ($24 \leq X < 31$) | 68.57% |
| | D | 23 | 54.0 | Average ($19 \leq X < 24$) | 65.71% |
| | E | 18 | 43.3 | Below average ($14 \leq X < 19$) | 51.43% |
| | F | 23 | 54.0 | Average ($19 \leq X < 24$) | 65.71% |
| MMM | A | 53 | 67.1 | Above average ($44 \leq X < 54$) | 81.54% |
| | B | 53 | 67.1 | Above average ($44 \leq X < 54$) | 81.54% |
| | C | 43 | 55.9 | Average ($34 \leq X < 44$) | 66.15% |
| | D | 49 | 62.7 | Above average ($44 \leq X < 54$) | 75.38% |
| | E | 52 | 65.8 | Above average ($44 \leq X < 54$) | 80.00% |
| | F | 42 | 54.1 | Average ($34 \leq X < 44$) | 64.62% |

*4.3. Test of Applicative Efficacy of the Intelligence Assessment and Strategy Implementation System*

4.3.1. Analysis of the Overall Applicative Efficacy in the Region

After a short-term generation intervention, the results of the end-of-semester mathematics tests of 568 subjects, which were distributed across the second semester of the seventh grade (Results 1) and the first semester of the eighth grade (Results 2), were collected. After calculating the mean value of the results of the two rounds of examinations, it was found that over the short term, students' academic performance in mathematics slightly improved, with average marks increasing by 0.454. A further *t*-test on Result 1 and Result 2 revealed a statistically insignificant difference ($t = -1.154$, $p = 0.249 > 0.05$) despite a slight improvement in academic performance (see Table 8).

**Table 8.** T-test of Result 1 and Result 2.

|  | **Mean** | **N** | **Standard Deviation** | *t* | **df** | *p* |
|---|---|---|---|---|---|---|
| Result 1 | 80.870 | 568 | 15.593 | −1.154 | 567 | 0.249 |
| Result 2 | 81.324 | 568 | 17.827 | | | |

In addition to improvement in students' academic performance, it was learned from the leaders of five schools based in Hexi District and teachers of the subjects that participants of the assessment focused on five sub-dimensions where they had scored relatively low, that is, inspection, planning, knowledge about strategies, cognitive experience and affective experience, after referencing the results of their intelligence diagnosis and improvement strategies. Improving their mathematical metacognitive knowledge (MMK), mathematical metacognitive experience (MME) and mathematical metacognitive monitoring (MMM) levels helped increase their inner drive related to mathematical learning and enabled them to formulate long-term goals for mathematic studies, carry out self-evaluation in a correct way, effectively enhance their mathematical problem-solving ability and reduce their burdens of mathematical learning, thereby improving their learning efficiency and decreasing their mathematical learning burdens.

4.3.2. Analysis of Applicative Efficacy on Students Participating in the Case Study

After implementing targeted guidance interventions on individual students, we conducted a post-test on students' mathematics metacognitive levels and academic results. In combination with qualitative examinations of students' behavioral performance, descriptions of learning efficiency, self and mutual evaluation, the applicative efficacy of the intelligence assessment system were tested from both quantitative and qualitative perspectives.

(1)    Analysis of applicative efficacy on student A who had weak MMK

According to the comparative study, following the mathematics metacognitive intervention, student A obtained a post-test score of 124, while student D in the control group obtained a post-test score of 113, which means student A was 11 scores higher than student D, indicating that the mathematics cognitive level of the former had significantly increased; with respect to metacognitive knowledge, student A's score increased to 46, representing a 35.29% increase, while student D obtained a score of 38, representing only a 5.26% increase. In terms of individual specific changes after the intervention, student A's mathematics metacognition was at the medium level prior to the intervention and at the top level after the intervention; the same student's metacognitive knowledge was at the medium level prior to the intervention and at the top level following the intervention. Therefore, the results stated above indicate that the improvement strategy of mathematics metacognitive knowledge is effective and can achieve better results if it is implemented based on the intelligence strategy implementation plan.

We communicated with H, a teacher, about the student's behavioral performances and obtained the following results. With regard to classroom learning behaviors, H stated that student A had significantly reduced such behaviors as classroom distraction and inattentiveness and started to take notes of teachers' writing on the blackboard, showing

significantly improved classroom efficiency; he was able to actively participate in classroom discussions and raise his own questions, along with an increased number of classroom presentations; he had also developed a practice of underlining important information when solving mathematic problems, resulting in higher accuracy and problem-solving speed. With respect to autonomous learning, H stated that student A would preview new lessons, underline relevant concepts and imitate sample solutions to solve exercises, exhibiting an increasingly higher autonomous learning ability. As a result, the student's academic results in mathematics had also significantly improved from level B to A.

During our interview with the student about learning efficiency and burdens, student A mentioned, "I am now a very attentive student in the class, and I finish my homework very quickly. That gives me time to review lessons. I am not as anxious as I used to be and no longer afraid of learning mathematics". This indicates that the student has improved his learning efficiency, reduced his academic burdens and improved his learning efficacy in mathematics. Thus, improving students' mathematics metacognitive level helps them increase their learning efficiency and master how to study.

(2)    Analysis of applicative efficacy on student B who had weak MME

According to the comparative study, following the mathematics metacognitive intervention, student B obtained a post-test score of 123, while student E in the control group obtained a post-test score of 114, which means student B was 9 scores higher than student E, indicating that the mathematics cognitive level of the former had significantly increased; with respect to metacognitive experience, student B's score increased to 25, representing a 56.25% increase, while student E obtained a score of 19, representing only a 1-score increase. In terms of individual specific changes after the intervention, student B's mathematics metacognition was at the medium level prior to the intervention and at the top level after the intervention; the same student's metacognitive experience was at the low to medium level prior to the intervention and at the top level following the intervention. Therefore, the results stated above indicate that the improvement strategy of mathematics metacognitive experience is effective.

We communicated with H about the student's behavioral performances and obtained the following results. With regard to classroom learning behaviors, H stated that student B had gradually developed an interest in mathematical learning, become more attentive in class and bold enough to raise questions when he came across things he did not understand; he gradually caught up with the learning progress, was able to integrate himself in group discussions and showed increased correctness level in answering classroom questions; in exercise classes, he would try to overcome all obstacles and attempt different approaches to solving problems. With respect to autonomous learning, H stated that the student would preview new lessons, underline key and difficult points and occasionally imitate sample solutions to solve exercises, exhibiting an increasingly higher autonomous learning ability; he also started to try summarizing the knowledge and incorrectly answered questions for each chapter. The student's academic result level in mathematics was still B without significant changes; however, the ranking of the result changed significantly from 60−70% to somewhere around 50%.

During our interview with the student about learning efficiency and burdens, student B mentioned, "I have behaved very well recently, and my teacher praised me for maintaining a good state of learning. I no longer dislike learning mathematics". When asked about questions related to the efficiency of mathematical learning, he said that "I used to have much homework that I did not know how to solve, and it took me a long time to finish it. Now, all things have got better. Most importantly, I no longer struggle with doing my homework. I am able to better focus my energy, and there is still some time left after I finish my homework. It feels good". This indicates that student B had improved his learning efficiency, enhanced his confidence and reduced his academic burdens, with learning anxiety alleviated.

(3)    Analysis of applicative efficacy on student C who had weak MMM

According to the comparative study, following the mathematics metacognitive intervention, student C obtained a post-test score of 124, while student F in the control group obtained a post-test score of 113, which means student C was 11 scores higher than student F, indicating that the mathematics cognitive level of the former had significantly increased; with respect to metacognitive monitoring, student C's score increased to 56, representing a 30.23% increase, while student F obtained a score of 46, representing only a 9.5% increase. In terms of individual specific changes after the intervention, student C's mathematics metacognition was at the medium level prior to the intervention and at the top level after the intervention; the same student's metacognitive monitoring was at the medium level prior to the intervention and at the top level following the intervention. Across sub-dimensions, student C scored 9 and 12 on planning and inspection, representing a 50% and 100% increase, respectively, with the largest increase taking place in the inspection dimension. The student's regulation and evaluation dimensions had also gained steady growth, increasing by 13.3% and 33.3%, respectively, compared with those of the pre-test. The student's management dimension was maintained at a relatively high level. Therefore, the results stated above indicate that the improvement strategy of mathematics metacognitive monitoring is effective.

We communicated with H about the student's behavioral performances and obtained the following results. With regard to classroom learning behaviors, H stated that student C had reduced the number of inattentive behaviors in class, was able to monitor his speech and conduct and strove to catch up with the teacher's progress; he was able to reflect over or inspect his shortcomings and adjust himself to achieve an optimal state of study. With respect to autonomous learning, H stated that the student would formulate simple study plans and implement them; he had made great progress in terms of correcting his errors after class and recording the causes of these errors, which helped reduce the occurrence of repeated errors. Student C managed to elevate his ranking in mathematics performance, from a low to medium level, to a medium to high level.

During our interview with the student about learning efficiency and burdens, student C thought that he was able to autonomously regulate and control his speech and conduct; he had developed a habit of formulating study plans and completing various tasks in an organized fashion based on these plans; he was also able to sort out his mind first when solving problems and had significantly improved his study efficiency. Part of the interview is excerpted as follows:

Question: "What would you first do when you get a mathematical problem with medium difficulty?"

Student C: "I would first analyze the question, underline the keywords and mark what can be inferred from given conditions. Then, I would look at the conclusion part, and what I need to do in order to find the proof of the conclusion. This is a stepwise analytical process. After the analysis, I would basically have a plan for solving the problem".

To sum up, by applying the intelligence assessment system and based on assessment diagnosis and improvement strategies in the system, we implemented regional and individual specific intervention and improvement, followed by research adopting multiple approaches, such as tests, questionnaires, interviews and text analyses. The results show that the overall academic performance in mathematics in the researched region had significantly improved. The academic result level, metacognitive level and learning habits of students participating in the case study had invariably increased. Students' evaluations from themselves and teachers reflect that students' learning efficiency had also increased, along with their academic burdens lightened. Thus, the intelligence assessment system is highly efficient, convenient and effective.

## 5. Conclusions

A mathematics metacognitive intelligence assessment and strategy implementation system for junior high school students was developed. The system comprised the assessment structural model, assessment scale, regional norm, targeted improvement strategies

for students at different levels and the intelligence assessment and strategy implementation software incorporating the said content.

Through the application of the intelligence assessment and strategy implementation system in Hexi District, Tianjin, it can be accurately diagnosed that junior high school students in Hexi District have both advantages and shortcomings in mathematics metacognition; further, professional and personalized strategies for improvement were proposed based on scientific evaluation. For example, the accurate diagnosis reveals that the students in Hexi District have excellent mathematics metacognition and exhibit considerable advantages in mathematical metacognitive knowledge and mathematical metacognitive management but are relative and have much room for improvement in mathematical metacognitive experience, especially in affective experience. Following the intervention based on the improvement suggestions generated by the intelligence assessment and strategy implementation system, firstly, it showed that students' overall academic performance in mathematics had improved but without a statistically significant difference. Moreover, our interviews with teachers show that after the general intervention, students had significantly improved their learning behaviors and habits and cognitive experience in mathematics learning, which were specifically reflected by their ability to formulate learning targets, select learning approaches, develop problem-solving plans and learn autonomy and imitativeness. Additionally, the results of the intervention in case studies also showed that students' metacognition, academic results, learning behaviors and efficiency had all significantly improved, indicating that the improvement strategies devised in this research are of certain positive effect in increasing students' mathematics metacognition, thus demonstrating the effectiveness of the application of the mathematics metacognitive intelligence assessment and strategy implementation system among junior high students.

Combining both theoretical and practical experience, this research developed a mathematics metacognitive intelligence assessment and strategy implementation system for junior high school students. The system can not only fulfill autonomous diagnosis and strategy implementation by individual students, but it can also process bulk and large-scale big data of mathematics metacognitive assessment and propose targeted measures and suggestions for improvement based on diagnostic results of the assessment of each student. Compared with previous metacognitive research, firstly, the system improves the efficiency of mathematical metacognitive evaluation and can efficiently complete the evaluation and output the results. Secondly, the system incorporates the mathematical metacognition norm. The evaluation results include not only the mathematical metacognition score but also the relative position of the score in the city, which provides the evaluator with a more objective positioning of their own level.

In the era of big data, intelligent evaluation can effectively improve the efficiency of educational researchers, and the development of mathematical metacognitive intelligent evaluation system can provide imitation ideas and framework for evaluation research. However, the research also has some limitations. First, only the norm of Tianjin, China, was included in the system, and norms of different regions should be further developed and embedded into the system to enhance the universality of the system. In addition, only one region of China was selected for the application of the system, and it is still worth considering whether the effect of the system can be tested through the application of only one region. Further, the sample should be expanded to assess the system more comprehensively and to achieve continuous optimization.

**Author Contributions:** Conceptualization, G.W.; methodology, G.W. and Y.K.; software, G.W. and Y.K.; validation, Z.J., Y.Z. and D.Z.; formal analysis, Y.Z. and Z.J.; investigation, Y.K., D.Z. and M.S.; resources, G.W.; data curation, Y.Z. and Z.J.; writing—original draft preparation, Y.K. and Z.J.; writing—review and editing, X.C. and Y.Z.; visualization, Y.K. and M.S.; supervision, G.W.; project administration, G.W.; funding acquisition, G.W. All authors have read and agreed to the published version of the manuscript.

**Funding:** This research was funded by the Key Cultivation Project of Tianjin Teaching Achievement Award: Research and Development of Mathematics Learning Assessment Tool and Its Practical Application, grant number PYJJ-036.

**Institutional Review Board Statement:** Not applicable.

**Informed Consent Statement:** Informed consent was obtained from all subjects involved in the study.

**Data Availability Statement:** The data presented in this study are available from the corresponding author upon reasonable request.

**Acknowledgments:** We thank the reviewers who helped us improve the manuscript.

**Conflicts of Interest:** The authors declare no conflict of interest.

## Appendix A

*Questionnaire of Junior High School Students' Mathematics Metacognition Level*

There are 36 questions in the questionnaire, which are scored on a five-point Likert scale. The questionnaire includes 31 mathematical metacognitive questions and 5 lie detection questions. Mathematical metacognitive questions are designed to assess students' mathematical metacognitive levels, and lie detection questions are used to eliminate invalid questionnaires. The total score of the questionnaire is 155 (lie detection questions do not count toward the total score). The higher the student's total score, the higher the level of mathematics metacognition. In the test, students can choose to answer the questionnaire online or fill in the questionnaire with paper and pencil according to their situation. The formal questionnaire is as follows.

Dear students, this survey on mathematical learning aims to provide you with targeted suggestions that help you improve your mathematical attainment. Your score does not define how well you will learn mathematics. Rather, it only functions as an objective criterion to evaluate your mathematical learning conditions. Please answer the questions carefully.

Note: 5 = Absolutely appropriate, 4 = Appropriate, 3 = Uncertain, 2 = Inappropriate, 1 = Absolutely inappropriate.

**Table A1.** Questionnaire of Junior High School Students' Mathematics Metacognition Level.

| 1 | Mathematical Metacognitive Knowledge | Score | | | | |
|---|---|---|---|---|---|---|
| 1.1 | I am aware of my mathematical learning ability and confident that I can solve various mathematical questions on my own. | 5 | 4 | 3 | 2 | 1 |
| 1.2 | I have a relatively clear understanding of common solving or demonstration approaches to mathematical problems. | 5 | 4 | 3 | 2 | 1 |
| 1.3 | I am aware of the goals or tasks of my mathematical learning. | 5 | 4 | 3 | 2 | 1 |
| 1.4 | I know what knowledge is examined in mathematical homework assigned by my teacher. | 5 | 4 | 3 | 2 | 1 |
| 1.5 | I am able to grasp mathematical knowledge (e.g., concept, equation and theorem) taught in classroom. | 5 | 4 | 3 | 2 | 1 |
| 1.6 | I am able to discern the level of my understanding about some issues. | 5 | 4 | 3 | 2 | 1 |
| 1.7 | I know whether I have understood the mathematical content I learned. | 5 | 4 | 3 | 2 | 1 |
| 1.8 | I have a clear understanding of the types of mathematical learning tasks (e.g., autonomous learning and group discussion). | 5 | 4 | 3 | 2 | 1 |
| 1.9 | I often adopt a variety of methods to solve mathematical problems. | 5 | 4 | 3 | 2 | 1 |
| 1.10 | I find that I have actively used effective learning strategies. | 5 | 4 | 3 | 2 | 1 |
| 1.11 | I adopt different learning approaches for different mathematical content. | 5 | 4 | 3 | 2 | 1 |

**Table A1.** *Cont.*

| 2 | Mathematical Metacognitive Experience | Score | | | | |
|---|---|---|---|---|---|---|
| 2.1 * | I cannot connect newly learned mathematical concepts or theorems with similar knowledge (e.g., cannot connect linear equations in one unknown with linear inequalities in one unknown). * | 5 | 4 | 3 | 2 | 1 |
| 2.2 | I try to find out central ideas behind mathematical problems (e.g., overall substitution). | 5 | 4 | 3 | 2 | 1 |
| 2.3 | In mathematical learning, the times I make mistakes will be fewer if I noticed such mistakes many times. | 5 | 4 | 3 | 2 | 1 |
| 2.4 | I realize that I have to plan the goals of my mathematical learning. | 5 | 4 | 3 | 2 | 1 |
| 2.5 | Successfully solving mathematical problems makes me happy. | 5 | 4 | 3 | 2 | 1 |
| 2.6 | I have a sense of accomplishment after completing mathematical homework. | 5 | 4 | 3 | 2 | 1 |
| 2.7 * | I am always confident about myself before learning new mathematical knowledge. * | 5 | 4 | 3 | 2 | 1 |
| **3** | **Mathematical Metacognitive Monitoring** | **Score** | | | | |
| 3.1 | After solving a problem, I would carefully summarize the inherent connections between different knowledge points to deepen my understanding. | 5 | 4 | 3 | 2 | 1 |
| 3.2 | Before taking a mathematics test, I would review relevant content in a planned manner (e.g., knowledge points where errors can easily occur or content that has not been adequately acquired). | 5 | 4 | 3 | 2 | 1 |
| 3.3 | When a mathematical problem cannot be solved by one method, I would timely turn to other problem-solving strategies. | 5 | 4 | 3 | 2 | 1 |
| 3.4 | When solving problems, I frequently remind myself of the necessity to pay attention to the given conditions or conclusions. | 5 | 4 | 3 | 2 | 1 |
| 3.5 | When I meet difficulties, I would try to re-find the solutions. | 5 | 4 | 3 | 2 | 1 |
| 3.6 | If I do not understand a mathematical concept, I would analyze an actual example related to the concept. | 5 | 4 | 3 | 2 | 1 |
| 3.7 | In mathematical learning, I would reflect over areas I have not fully grasped. | 5 | 4 | 3 | 2 | 1 |
| 3.8 | After a certain period of mathematical learning, I would evaluate the effectiveness of my learning in various ways. | 5 | 4 | 3 | 2 | 1 |
| 3.9 | After solving a problem, I would check whether my method is correct. | 5 | 4 | 3 | 2 | 1 |
| 3.10 | When solving a mathematical problem, I would think of whether I have solved its key questions. | 5 | 4 | 3 | 2 | 1 |
| 3.11 | When I have finished my mathematical homework, I would repeat some of the key parts to ensure that I have fully understood them. | 5 | 4 | 3 | 2 | 1 |
| 3.12 | I would memorize some problem-solving techniques (e.g., when doing an operation, start with involution, followed by multiplication and division and finally addition and subtraction). | 5 | 4 | 3 | 2 | 1 |
| 3.13 | I can better understand a problem when I take notes of its knowledge points. | 5 | 4 | 3 | 2 | 1 |
| **4** | **Lie Detection Questions** | **Score** | | | | |
| 4.1 | In mathematical learning, I would reflect over areas I have not fully grasped. | 5 | 4 | 3 | 2 | 1 |
| 4.2 | I am aware of the goals or tasks of my mathematical learning. | 5 | 4 | 3 | 2 | 1 |
| 4.3 | When solving a mathematical problem, I would think of whether I have solved its key questions. | 5 | 4 | 3 | 2 | 1 |
| 4.4 | Successfully solving mathematical problems makes me happy. | 5 | 4 | 3 | 2 | 1 |
| 4.5 | I never erroneously solve a mathematical problem. | 5 | 4 | 3 | 2 | 1 |

Note: * means reverse scoring questions.

## Appendix B

**Table A2.** Initial construction of improvement strategy for mathematical metacognition of junior high school students.

| Performance | Improvement Strategy |
|---|---|
| Dimension: Mathematical metacognitive knowledge (55 points) | |
| Sub-dimension: Knowledge about individuals (25 points) | |
| Middle-level students ($13 \leq X < 17$): <br> 1. There is room for improving the understanding of their own ability and effort level; <br> 2. Perform average in understanding their own knowledge mastery and problem-solving ideas; <br> 3. There is room for improving the understanding of their own mathematical learning characteristics. <br><br> Low-level students ($X < 13$): <br> 1. The understanding of their own ability and effort level is deficient; <br> 2. Perform poorly in understanding their own knowledge mastery and problem-solving ideas; <br> 3. There is a poor understanding of their own mathematical learning characteristics. | 1. Regularly reflect on the situation of mathematical learning and analyze the causes <br><br> Students should regularly reflect on their own grasp of mathematical knowledge and completion of exercises, analyze the reasons for the poor grasp of knowledge or mistakes in exercises, classify the reasons into personal ability, effort level, the difficulty level of task, etc., and treat their own ability and effort level more objectively through reflection. <br><br> 2. Summarize and review the mathematical knowledge frame of each chapter <br><br> Summarize the knowledge structure of each chapter (including theorems, definitions and connections between knowledge points) and review it in time. During the review, they should recall the knowledge points and the solving ideas of error-prone questions by the knowledge frame, mark the forgotten contents and fully understand their own advantages and disadvantages in knowledge and problem solving. <br><br> 3. Correctly understand the differences in mathematical learning between themselves and their peers <br><br> Communicate with peers to share their own mathematical learning (how to preview, take class, review, etc.), compare the similarities and differences with peers, and keep up common and good characteristics; actively improve their own mathematical learning according to a personal situation. |
| Sub-dimension: Knowledge about tasks (15 points) | |
| Middle-level students ($8 \leq X < 10$): <br> 1. The understanding of the nature and requirements of mathematical problems remains to be improved; <br> 2. There is room for improving the understanding of the types of mathematical tasks and the time needed to complete tasks; <br> 3. Perform average in the application of mathematical knowledge. <br><br> Low-level students ($X < 8$): <br> 1. Have a poor understanding of the nature and requirements of mathematical problems; <br> 2. There are deficiencies in understanding the types of mathematical tasks and the time required to complete tasks; <br> 3. Perform poorly in the application of mathematical knowledge. | 1. Remind oneself in the process of solving problems <br><br> Students should remind themselves to consider the error-prone points before solving a problem. For example, facing an equation solving problem, one should consider the denominator is not zero when the fraction value is zero. Additionally, remind oneself to think about the knowledge examined by the problem and the skills used to solve the problem after solving the problem to comprehensively deepen personal understanding of the question. <br><br> 2. Scientifically analyze and plan mathematical tasks <br><br> First, for multiple mathematical tasks, students should analyze whether the purpose is to summarize knowledge, consolidate practice methods or improve mathematical ability, determine the task types and complete them one by one from easy to difficult; second, they should estimate the time to complete each task, work out a good plan and strengthen their awareness of efficiency. <br><br> 3. Use equations to grasp key features of mathematical objects <br><br> Summarize the application of the same knowledge point in questions about variants, appropriately find some analytical cases to help oneself pay attention to the new characteristics of the same knowledge to promote one's better understanding of knowledge and better transfer and application of knowledge. |

**Table A2.** *Cont.*

| Performance | Improvement Strategy |
|---|---|
| **Sub-dimension: Knowledge about strategies (15 points)** | |
| Middle-level students (8 ≤ *X* < 10):<br><br>1. Perform average in reflecting on the use of mathematical learning strategies;<br>2. There is room for improvement in choosing mathematical ideas and methods;<br>3. Perform average in judging the advantages and disadvantages of problem-solving methods.<br><br>Low-level students (*X* < 8):<br><br>1. Perform poorly in reflecting on the use of mathematical learning strategies;<br>2. Are weak in choosing mathematical ideas and methods;<br>3. Are weak in judging the advantages and disadvantages of problem-solving methods. | 1. Reflect on the thinking process and strengthen the study of problem-solving strategies<br><br>Recall the thinking process and reflect on how to read key information, how to find the relationship between known conditions and unknown quantities, how to clearly and rigorously express the problem-solving process and even how to identify the sources of errors when setbacks are encountered, how to seek new problem-solving strategies to improve students' understanding about the use of these strategies.<br><br>2. Summarize mathematical ideas and methods<br><br>In the process of learning concepts and theorems, students should pay attention to the mathematical ideas and methods (conversion, symbolic–graphic combination, structure, etc.) and summarize by themselves every week; in the problem-solving process, they should first identify whether the problem situation is similar, then recall the relevant knowledge points and the mathematical ideas and methods that may be used and remind themselves to use the corresponding strategies.<br><br>3. Employ an explicit thinking process<br><br>Answer questions actively in class and broaden one's methods of solving problems by dictating or writing down the solving process and thinking about others' solving methods. Explain the problem to each other after class and summarize questions involving "one question solvable by multiple methods or one method capable of solving multiple questions", as well as methodologies. |
| **Dimension: Mathematical metacognitive experience (35 points)** | |
| **Sub-dimension: Cognitive experience (20 points)** | |
| Middle-level students (11 ≤ *X* < 14):<br><br>1. There is room for improvement in understanding the difficulty levels and familiarity levels of learned mathematical contents;<br>2. Perform average in experiencing and overcoming mathematical learning difficulties.<br><br>Low-level students (*X* < 11):<br><br>1. Are weak in understanding the difficulty levels and familiarity levels of learned mathematical contents;<br>2. There are deficiencies in experiencing and overcoming mathematical learning difficulties. | 1. Preview before class to enhance students' experience in the levels of learning content difficulty and familiarity<br><br>Previewing methods for different learning content are as follows. In the preview of mathematical knowledge about numbers and algebra, students are encouraged to make bold attempts, independently analyze questions and solve sample questions to build up knowledge of the methods. In the preview of mathematical knowledge about spatial patterns, students are encouraged to experience and perceive new knowledge through hands-on practices, such as cutting, joining, folding, measuring, etc. In the preview of mathematical knowledge about statistics and probability, students are encouraged to collect information via various channels, such as the internet, and try to understand and analyze new knowledge based on existing knowledge and experience.<br><br>2. Reflect on the ways of overcoming learning difficulties<br><br>Mathematical learning obstacles can be divided into the following types: inadequacy in logical thinking, spatial thinking and calculation abilities. To overcome these obstacles, it is imperative to use a number of approaches, including relating new knowledge with old knowledge, utilizing physical models and the combination of numbers and shapes and seeking simpler calculation methods. While solving problems and learning knowledge, students are encouraged to reflect on the types of their mathematical learning obstacles and try to overcome them. |

**Table A2.** *Cont.*

| Performance | Improvement Strategy |
|---|---|
| Sub-dimension: Affective experience (15 points) | |
| Middle-level students ($8 \leq X < 10$):<br>1. There is room for improvement in experiencing a sense of achievement in mathematics learning;<br>2. The confidence in mathematical learning remains to be improved.<br>Low-level students ($X < 8$):<br>1. Seldom experience a sense of achievement in mathematics learning;<br>2. Lack confidence in mathematical learning. | 1. Emphasize the comparison of cooperative and individualistic<br>Students attach great importance to group honor, actively participate in group activities to help their groups obtain praise and experience the joy of success through inter-group comparison; students compare their past mathematical learning state and result with present ones to constantly outdo themselves instead of overlooking their knowledge foundations, and they also conduct horizontal comparisons between classmates to experience the joy of constant improvement.<br>2. Change one's passive attitude toward homework<br>Treat homework correctly, change the passive attitude of "report on one's mission to the teacher", regard homework as a tool to improve the degree of their understanding of knowledge and the application of problem-solving strategies, ask questions proactively and summarize knowledge points or methods to experience the joy of gaining knowledge in the process of mastering and applying knowledge.<br>3. Give oneself positive psychological hints<br>Students often motivate themselves and are adept at finding their advantages in mathematical learning, such as thinking agility, the ability to identify problems and the courage to question authority; they are not indulged in negative information or sentiments and are willing to solve problems by using their hands, heads or turning to other classmates, thereby increasing their confidence. |
| Dimension: Mathematical metacognitive monitoring (65 points) | |
| Sub-dimension: Planning (10 points) | |
| Middle-level students ($5 \leq X < 7$):<br>There is room for improvement in working out mathematical learning plans.<br>Low-level students ($X < 5$):<br>Are weak in working out mathematical learning plans. | 1. Prepare daily to-do lists and set long-term goals<br>Prepare a to-do list every day, including daily tasks and their corresponding time quantum, strictly implement the list, avoid delaying or forgetting any task. Develop long-term mathematical learning goals and decompose them into more operational near-term goals, such as "my weekly test result must exceed that of xxx classmate" to motivate oneself to complete learning tasks.<br>2. Clarify problem-solving plans<br>Before answering a question, fully understand the meaning of the question and mark the key information; in the process of solving a problem, find out the implied conditions, formulate a solving plan and reversely infer from the target; afterward, reflect on whether there are areas where errors are easily made or whether there are spelling mistakes after solving a problem.<br>3. Work out a pre-examination review plan<br>Before an exam, review the knowledge points of the corresponding chapters according to the scope of examination, look for one's defects on those knowledge points and the reasons for mistakes. Prepare a list and delete one item upon overcoming it until all items are settled; solve key problems that are most likely to take effect in a short term specific to one's reality. |

**Table A2.** *Cont.*

| Performance | Improvement Strategy |
|---|---|
| **Sub-dimension: Regulation (20 points)** | |
| Middle-level students ($11 \leq X < 14$):<br>1. There is room for improvement in adjusting problem-solving methods;<br>2. Perform average in adjusting the attitude to mathematics learning;<br>3. Perform average in adjusting learning steps and plans.<br><br>Low-level students ($X < 11$):<br>1. Perform poorly in adjusting problem-solving methods;<br>2. Are weak in adjusting the attitude to mathematics learning;<br>3. Are weak in adjusting learning steps and planning. | 1. Pay attention to problem-solving techniques without stubbornly sticking to one method only<br><br>If there is more than one idea, but one is sure about none, one should encourage oneself to explore each idea a little bit, apply heuristic self-questioning, such as " What features do the questions have? What are the conditions? What is the conclusion? ". One can adjust one's thinking process and problem-solving methods through self-awareness.<br><br>2. Adjust one's mentality in time<br><br>Treat mathematical learning with a usual mind. When one makes progress or is praised, one should reflect on one's own drawbacks and regulate one's self-contented or proud emotions; when one falls behind or is criticized, one can talk with peers, seek help and relieve one's frustration.<br><br>3. Adjust mathematics learning steps and plans in time<br><br>Reflect on one's own mathematical learning situation regularly to find one's own shortcomings, timely adjust the learning plan to make up for the shortcomings, such as enhancing mastery through practice. |
| **Sub-dimension: Evaluation (10 points)** | |
| Middle-level students ($5 \leq X < 7$):<br>1. There is room for improvement in evaluating their own mastery of classroom content;<br>2. Do average in evaluating their own problem-solving thinking.<br><br>Low-level students ($X < 5$):<br>1. Are weak in evaluating their own mastery of classroom content;<br>2. Are weak in evaluating their own problem-solving thinking. | 1. Actively carry out self-evaluation over their grasp of classroom content<br><br>Actively use "a question chain" to evaluate one's own knowledge mastery., e.g., "What are the knowledge points involved in this class? Which have been mastered? Which have not? Additionally, why? What are the countermeasures?". Students change passive thinking (e.g., thinking that post-learning summarization is the task of the textbook or teachers).<br><br>2. Pay attention to reflective evaluation on problems involving recurring errors<br><br>Strive to find in-depth causes of recurring errors, including inadequate knowledge, careless analysis, unclear thinking, wrong solving approach, missing solutions and calculation problems; organize erroneous questions by categories and review them on a weekly basis; practice variant exercises until no similar mistakes are made.<br><br>3. Write reflective weekly reports on mathematics<br><br>Comprehensively reflect on the thinking process of problem solving, conduct multi-angle (problem-solving ideas, problem-solving ways, etc.) observation according to the structural characteristics of problems and write weekly reports on the problem-solving methods acquired through reflection. In the future study, apply the acquired problem-solving methods, reflect on the problems existing in the application, seek ways to overcome, supplement and improve reflective weekly reports to gradually develop the habit of reflection–remediation–reflection. |

**Table A2.** *Cont.*

| Performance | Improvement Strategy |
|---|---|
| Sub-dimension: Inspection (15 points) | |
| Middle-level students ($8 \leq X < 11$):<br>1. There is room for improvement in testing their own learning results;<br>2. Do average in judging the rationality of their chosen problem-solving strategy.<br>Low-level students ($X < 8$):<br>1. Perform poorly in testing their own learning results;<br>2. Are weak in judging the rationality of their chosen problem-solving strategy. | 1. Mark task lists to test one's own learning results<br>Students mark daily or weekly task lists (divided by star levels), check and evaluate each other with peers and exchange and share experience; complete after-class exercises in the textbook or related exercises after each class to understand their own knowledge application situation.<br>2. Review the problem-solving process, summarize the key and verify it<br>After solving a problem, students reflect on how they identify and solve the key problems, what basic thinking methods, skills and techniques they have applied, what detours they have taken, what are the areas where errors can easily occur (or have occurred), what are the causes, what lessons can be drawn; find similar problems to verify the universality of their problem-solving method.<br>3. Self-test to judge the rationality of strategies<br>After solving the problems, reflect on the effectiveness of the strategies used and look for similar problems to solve to test the generalizability of the solution strategies. If problems are found in the use of the solution strategy, find the reasons and avoid the unreasonable use of the solution strategy. |
| Sub-dimension: Management (10 points) | |
| Middle-level students ($5 \leq X < 7$):<br>1. There is room for improvement in inducing mathematical knowledge;<br>2. Do average in summarizing mathematical problem-solving methods.<br>Low-level students ($X < 5$):<br>1. Are weak in inducing mathematical knowledge;<br>2. Are weak in summarizing mathematical problem-solving methods. | 1. Take notes and sort out mistakes<br>Quickly take down the teacher's important blackboard writing and supplementary knowledge in class and complete the notes after class. Sort out one's mistakes according to knowledge points and specify the causes of mistakes, the background knowledge involved and the problem-solving methods used.<br>2. Summarize problem-solving techniques and steps by category<br>For commonly seen questions, students summarize the general methods and techniques for problem solving, such as the conversion approach commonly used in geometry, the fixed steps in solving quadratic equations and the "let-but-not-solve" strategy in solving values of algebraic equations. |

Note: Above middle-level are high-level students, and the improvement strategy is "Keep up".

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
