# Peer review of "Development and Application of Intelligent Assessment System for Metacognition in Learning Mathematics among Junior High School Students"

_sustainability, doi:10.3390/su14106278_

Round 1
Reviewer 1 Report
The article is devoted to the development of a strategy for the formation of metacognitive competence in middle-aged schoolchildren. An intellectual system for assessing metacognitive skills has been developed. An experimental verification of the correctness of the strategy showed the effectiveness of the proposed methods. The studies were carried out at a high level, well substantiated. The text of the article is logical, well structured and understandable. I have no complaints about the article.
Author Response
Dear Reviewer,
Thank you for your support of our research!
We continued to revise the manuscript, mainly in the introduction, literature review, instruments and conclusions sections.
Thank you for your careful review. We really appreciate your efforts in reviewing our manuscript during this unprecedented and challenging time. We thank you and remain at your disposal for any further questions.
Reviewer 2 Report
I do not think that the text presented has anything to do with the focus of the magazine. This fact is also due to the fact that the authors do not work with the concept of sustainability anywhere. If the editor decides otherwise, I continue to describe the review of the whole text in the points that I recommend the authors to incorporate (provided that the editor judges the suitability of the whole text)
- List of literature. In many cases, this is very outdated literature. Author Wang, G. In addition, he cites himself a total of 4 times, which of course will lead to an increase in citations to WoS. In addition, these citations are included in only one paragraph in the text, which I believe may be absent. From this initiative, it follows that it is necessary to redesign part of the theory so that it is drawn from current sources.
- As one of the main variables monitored is mathematical metacognition, it considers it necessary to present the whole tool in the form of an appendix and describe how many points the respondent can get, how respondents are evaluated, etc. Many tests work for comparison with experts /www.mdpi.com/2227-7390/8/6/969/htm)
- If I am not mistaken, the test "Test of academic performance in mathematics" was selected on the basis of an interview with Mr. Wang. Has this test been piloted? I think it is necessary to mention the item analysis, including the ULI coefficient (sensitivity, difficulty, etc.). This part needs to be added to the text.
- Images are inserted into the text as part of the visualization. However, the text is in Chinese and not in English. If the picture is to be telling to the reader, the sample must be done in English. This applies to all images.
- Notation p = 0.000 <0.01) is quite non-standard.
Reviewer 3 Report
This is an interesting paper about the design and evaluation of a system to support metacognition in mathematics education. My sense is that the underlying work is strong, and in technical terms the research appears to be on firm ground. Yet the paper is uneven, with some aspects of the design and the argument of the paper needing to be expanded to help the reader understand the system and its implications. I therefore think the authors should be required to address a number of issues before publication is considered:
- The core argument of the paper is expressed most directly on line 142-149, which is far too late and expressed too briefly. That argument needs to be (a) moved earlier into the Abstract and Introduction and expanded and (b) registered more forcefully throughout the literature review.
- The Abstract needs to be much more explicit about the most important findings and what they add to the current literature on the topic.
- We really need a dedicated Introduction section that sets out (a) the key argument of the paper, (b) the motivation for carrying out the work, and (c) the structure of the rest of the paper. In the present version of the manuscript, the start of the paper is much too abrupt.
- The three literature review areas (numbered 1.1, 1.2 and 1.3) need to more explicitly establish what criticism the authors have of the works being reviewed, and what shortcoming in that literature the present work addresses. The current literature review established what is known and makes it seem like every issue is decided and closed rather than warranting further investigation.
- Where instruments are selected for use (e.g., lines 242 and 261) we need to know WHY they were selected (what made them appropriate in terms of the research objectives), rather than a simple statement that they were selected.
- When describing the system development, I think we need to see more clearly the earlier system before the changes described on pages 11-13 were made. Respondents make a large number of suggestions and recommendations, but we only see the system AFTER those changes have been considered. I think we need to see the earlier system to which they were responding, to help us contextualise their comments.
- When presenting Figures that illustrate the system, I think that we need to key elements labelled in English.
- Section 5 needs to more explicitly describe the contribution in terms of how the findings add something that will be of interest to the scholars whose work was reviewed in the three earlier literature review sections.
- There are several places (lines 339, 354, 381, 614, 648, 685) where the numbering of headings becomes confusing. I think that all heading numbering should be incorporated into the wider numbering structure.
Round 2
Reviewer 2 Report
Thank you very much for the opportunity to reconsider the article. I will comment gradually on all the points I criticized last time:
List of literature. In many cases, this is very outdated literature. Author Wang, G. In addition, he cites himself a total of 4 times, which of course will lead to an increase in citations to WoS. In addition, these citations are included in only one paragraph in the text, which I believe may be absent. From this initiative, it follows that it is necessary to redesign part of the theory so that it is drawn from current sources.
The introductory text was largely redesigned as required. However, it is still true that there is a high level of autovigation. I suspect Wang G. is currently quoted five times. The authors did not comment on this remark.
As one of the main variables monitored is mathematical metacognition, it considers it necessary to present the whole tool in the form of an appendix and describe how many points the respondent can get, how respondents are evaluated, etc. Many tests work for comparison with experts /www.mdpi.com/2227-7390/8/6/969/htm)
The note has been incorporated just like any other. After deleting the self-citations, the text can be published.
